# Spider 2.0: Evaluating Language Models on Real-World Enterprise Text-to-SQL Workflows

**Fangyu Lei**[*][♠] **Jixuan Chen**[*][♠] **Yuxiao Ye**[♠] **Ruisheng Cao**[♠] **Dongchan Shin**[♠]
**Hongjin Su**[♠] **Zhaoqing Suo**[♠] **Hongcheng Gao**[♠] **Wenjing Hu**[♠] **Pengcheng Yin**[♡]
**Victor Zhong**[★] **Caiming Xiong**[◇] **Ruoxi Sun**[△] **Qian Liu**[♣] **Sida I. Wang**  **Tao Yu**[♠]
[♠]University of Hong Kong    [◇]Salesforce Research    [♣]Sea AI Lab
[♡]Google Deepmind    [△]Google Cloud AI Research [★]University of Waterloo

## Abstract

Real-world enterprise text-to-SQL workflows often involve complex cloud or local data across various database systems, multiple SQL queries in various dialects, and diverse operations from data transformation to analytics. We introduce Spider 2.0, an evaluation framework comprising 632 real-world text-to-SQL workflow problems derived from enterprise-level database use cases. The databases in Spider 2.0 are sourced from real data applications, often containing over 1,000 columns and stored in local or cloud database systems such as BigQuery and Snowflake. We show that solving problems in Spider 2.0 frequently requires understanding and searching through database metadata, dialect documentation, and even project-level codebases. This challenge calls for models to interact with complex SQL workflow environments, process extremely long contexts, perform intricate reasoning, and generate multiple SQL queries with diverse operations, often exceeding 100 lines, which goes far beyond traditional text-to-SQL challenges. Our evaluations indicate that based on o1-preview, our code agent framework successfully solves only 21.3% of the tasks, compared with 91.2% on Spider 1.0 and 73.0% on Bird. Our results on Spider 2.0 show that while language models have demonstrated remarkable performance in code generation — especially in prior text-to-SQL benchmarks — they require significant improvement in order to achieve adequate performance for real-world enterprise usage. Progress on Spider 2.0 represents crucial steps towards developing intelligent, autonomous, code agents for real-world enterprise settings. Our code, baseline models, and data are available at `spider2-sql.github.io`.

## 1 Introduction

Automated code generation can serve as a crucial bridge between humans and data, assisting individuals in achieving difficult or monotonous tasks using complex data. A significant portion of existing data is stored in relational databases, where SQL serves as an essential interface that facilitates human interaction with these data. In this context, semantic parsing or text-to-SQL (Dahl et al., 1994; Zelle & Mooney, 1996; Zettlemoyer & Collins, 2005; Li & Jagadish, 2014; Zhong et al., 2017; Yu et al., 2018) is an important technology that assists data analysts in performing routine queries, orchestrating data workflows, and accomplishing advanced business intelligence, thereby significantly reducing repetitive human labor and alleviating the burden on programmers. Large language models (LLMs) have demonstrated excellent capabilities in generating code (Chen et al., 2021; Austin et al., 2021), particularly in transforming natural language questions into SQL queries. Notably, methods based on GPT-4 achieved execution accuracy of 91.2% and 73.0% on the classic benchmarks Spider 1.0 (Yu et al., 2018) and Bird (Li et al., 2024b), respectively.

Although LLMs excel on these datasets, they often use non-industrial databases with few tables and columns, featuring simplistic SQL and questions that fall short of real-world complexity and overlook diverse SQL dialects. By contrast, real-world data are stored across a diverse array of

---

[*]Equal contribution.

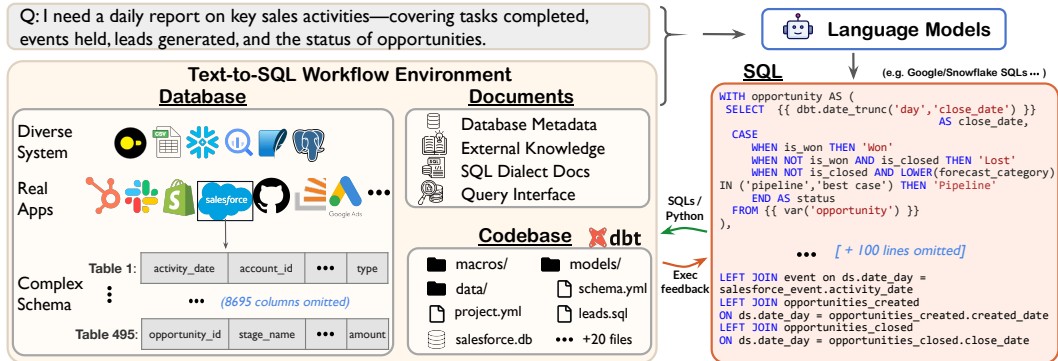

Figure 1: Spider 2.0 aims to evaluate LLMs on real-world enterprise-level text-to-SQL workflows. Solving each task requires understanding database metadata, consulting SQL dialect documentation, handling complex workflows, and performing intricate reasoning to generate diverse SQL queries.

database systems, each with its own unique SQL dialects, introducing a wide range of SQL syntax and functions. Additionally, these enterprise-level application databases are characterized by large-scale schemas with thousands of columns and complex nested structures. Moreover, real-world text-to-SQL workflows require the utilization of project codebases, external knowledge, and various contexts to construct intricate SQL queries across multiple steps, complete various operations, and build a comprehensive data engineering pipeline. This includes data wrangling to clean and organize the data for analysis, data transformation to restructure and enhance the data, and conducting data analytics to extract insights that inform decision making and drive strategic initiatives. All these complexities underscore the pressing need for a more realistic enterprise-level benchmark.

We present Spider 2.0, a benchmark that reflects real-world data workflows to facilitate the development of text-to-SQL models in enterprise applications, encompassing 632 real-world complex data wrangling, transformation, and analysis tasks. As illustrated in Fig. 1, the databases in Spider 2.0 are sourced from industrial applications (e.g. Google Analytics and Salesforce) and feature massive schema items (an average of 812 columns) with unique structures (e.g., nested columns in Fig. 11, multiple schema in Fig. 12), along with terabyte-scale data volumes. They encompass a variety of database systems, including local databases (e.g., SQLite and DuckDB) and cloud data warehouses (e.g., BigQuery and Snowflake). Complicated SQL dialects for these databases are curated from technical tutorials, community forums, and open-source projects. On average, each ground-truth SQL query contains 144 tokens and includes advanced functions (e.g., $\texttt{ST\_DISTANCE}(x_1, x_2)$ measures the shortest distance between two points), exhibiting a level of complexity notably surpassing previous benchmarks. All tasks are based on project codebases along with documents and database interface to simulate real-world text-to-SQL writing scenarios.

Unlike previous datasets, Spider 2.0's agentic task setting does not rely on pre-prepared inputs (question and database schema) or expected outputs (predicted SQL). Instead, it incorporates a real project codebase and a database interface. This complexity extends beyond merely predicting an SQL query; it involves navigating the project and dynamically interacting with complex databases through SQL queries and command-line scripts (in Python or Shell). The task objective is to perform intricate data transformations within the database or to extract analytical insights from the data. This task setting closely mirrors real-world enterprise SQL workflows, requiring the model to refer to the codebase and documentation, generate multiple SQL queries, and dynamically interact with the environment to complete complex tasks and derive the final result. To simplify performance comparisons with previous text-to-SQL methods and benchmarks, and to support faster development and evaluation, we also introduce Spider 2.0-lite and Spider 2.0-snow, self-contained datasets with preprocessed database schema and documentation, the former is hosted on BigQuery, Snowflake, and SQLite, while the latter is entirely hosted on Snowflake. This setting omits the codebase and restricts output to SQL only, thus eliminating the need to predict final answers or transform the database. While they are sourced from the same raw data as Spider 2.0, these settings are not easier than Spider 2.0 because text-to-SQL setting have access to less information (e.g., execution feedback). We present Spider 2.0-lite and Spider 2.0-snow as direct text-input-to-SQL-output challenges that are easier to work with using current advanced text-to-SQL parsers, and Spider 2.0 as

a real-world data workflow challenge that involves interacting with diverse sources to perform data transformation and analyses.

Our evaluation on Spider 2.0 indicates significant room for improvement in deploying LLMs within real-world enterprise text-to-SQL workflows. The best o1-preview based code agent framework achieves a performance of only 21.3%, underscoring the significant deficiency in LLMs' capability to serve as proficient SQL experts (Tab. 2). As for Spider 2.0-lite setting, even the most advanced text-to-SQL parser could successfully address only 5.7% of the questions, a stark contrast to the execution accuracy of 91.2% on Spider 1.0 and 73.0% on BIRD, thereby highlighting the substantial challenges (§3.2). Our detailed analysis further identifies major obstacles in enterprise text-to-SQL, including accurately linking schemas from extremely large databases, correctly handling different SQL dialects, planning sequences of nested SQL queries to perform complex transformation and analytical tasks, and effectively leveraging external documentations and understanding project-level codebase. (§4.1 and §4.2). These challenges in Spider 2.0 represent crucial steps toward creating a benchmark that closely aligns with real-world scenarios. With Spider 2.0, we aim to enable the development of a new generation of intelligent autonomous agents capable of data engineering workflows in real-world enterprise settings.

## 2 BENCHMARK CONSTRUCTION

In this section, we introduce the task definition, general annotation pipeline, and dataset statistics for Spider 2.0, Spider 2.0-snow and Spider 2.0-lite. For concrete examples, refer to App.B.

### 2.1 TASK DEFINITION

Fig. 2 illustrates the task definition of both code agent setting and traditional text-to-SQL setting.

**Code agent task.** Spider 2.0 is defined as a comprehensive code agent task. Given a question $\mathcal{Q}$, a database interface $\mathcal{I}$, and a codebase $\mathcal{C}$ (with project context, configuration, and documentation, illustrated in Fig. 1), the task is to iteratively modify the code (SQL/Python) $\mathcal{C}$ based on observations $O_k = \text{execute}(\mathcal{C}, \mathcal{I}, \mathcal{Q})$ until the final result $\mathcal{A}$ (text/table/database) is obtained. In other words, we use the final observation $O_k$ as an agent's answer to the question, i.e., $\mathcal{A} = O_k$.

**Text-to-SQL task.** In contrast to Spider 2.0, Spider 2.0-snow and Spider 2.0-lite are formulated as self-contained tasks. Given database schema $\mathcal{D}$, a natural language question $\mathcal{Q}$, and auxiliary documentation $\mathcal{E}$ as inputs, the text-to-SQL parser $f(\cdot)$ is required to output the corresponding SQL query $\mathcal{S} = f(\mathcal{Q}, \mathcal{D}, \mathcal{E} \mid \theta)$, where $\theta$ is the param-

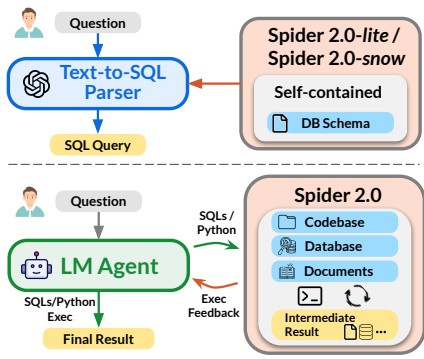

Figure 2: We offer two settings: traditional text-input-to-SQL-output Spider 2.0-lite/snow, and agentic Spider 2.0.

eters of the parser. Spider 2.0-lite's database is hosted on diverse databases like Spider 2.0, while Spider 2.0-snow is entirely hosted on Snowflake, with a greater focus on text-to-SQL generation.

### 2.2 ANNOTATION PIPELINE

Eight authors majoring in computer science, all highly proficient in SQL, carry out the data annotation process. The annotation pipelines consist of the following six steps:

**1) Database and SQL collection.** We collect various databases from cloud data warehouses, including BigQuery public data, Snowflake Marketplace data, and other platforms, to ensure that they meet specific criteria: each database must contain more than 200 columns or have a nested schema structure. After filtering, we select 74 BigQuery, 54 Snowflake, 30 SQLite, 40 DuckDB, 10 PostgreSQL, and 5 ClickHouse databases. From the corresponding tutorials and forums, we gather $1,021$ complex SQL queries, as well as 157 data transformation projects sourced from Fivetran and DBT (see App.B.2). To meet our criteria, the SQL queries must contain more than 50 tokens (tokenized by whitespace; for reference, the average token count of BIRD (Li et al., 2024b) is 30.9). Furthermore, queries must originate from real projects or tutorials, not from synthetic examples or corner cases. Ultimately, we retain 547 high-quality SQL queries and 78 DBT projects.

**2) SQL rewrite to prevent data leakage.** To avoid contamination and ensure the credibility of Spider 2.0's evaluation, annotators are required to rewrite each SQL and verify that they are bug-free. The rewrites are performed at two levels of increasing complexity: the surface and semantic levels, as detailed in Tab. 1. 84.2% of the examples underwent surface-level rewriting, while 42% experienced semantic-level rewriting. Annotators must ensure that the rewritten SQL executes successfully, completes in an acceptable time, and returns non-empty results. 85.98% of these SQL queries utilize advanced functions in various dialects (App.B.7.1), while 10.76% require additional DBT tools, posing challenges due to the need to integrate the project context.

Table 1: The rewrite categories are as follows: "Surface" rewrites adjust the parameters and the answer format, while "Semantic" rewrites expand the question's meaning. Each table reference in *Example* column represents the details of rewrite examples for the corresponding type.

| Rewrite | Categories | Example |
|---|---|---|
| Surface | Answer format | Tab. 13, replace the one channel with the channel ranking by sessions. |
| | Condition parameters | Tab. 14, more complex filter condition: citibike is faster than a taxi. |
| | Advanced calculation | Tab. 15, calculate originality score based on selected publications. |
| Semantic | Advanced requirements | Tab. 16, change *page view order* to *page conversion rate*. |
| | Merge related SQLs | Tab. 17, merge geography-related and weather-related queries. |
| | SQL codebase files | App.B.2, change SQL and YML files in the original project. |

**3) Codebase and context setup.** For each complex SQL query in Spider 2.0-lite and Spider 2.0-snow, we collect the external reference documents necessary to complete the task. Since the tasks span multiple database types, we gather documentation on SQL dialects and external functions, as shown in Tab. 18. Additionally, for Spider 2.0, we preserve the original codebase of the SQL-related project. For Spider 2.0, besides collecting reference documents, annotators also gather resources such as codebases, database interfaces to establish the context for each task (Fig. 1). Since some complex data transformation intentions may not be fully captured by a natural language question, annotators provide additional context, including data model descriptions (App.B.2) or predefined answer files (App.B.5), to maintain clarity while addressing potential ambiguities.

**4) Natural language task instructions annotation.** Annotators are required to write questions based on the SQLs and context gathered in Step 3, crafting two versions for different settings. The instructions are designed to balance both *naturalness* and *unambiguity*. Due to the differences between Spider 2.0 and Spider 2.0-lite/snow, code agent tasks demonstrate greater naturalness in its questions because it provides contexts and predefined files to guide the answers, while text-to-SQL tasks prioritize unambiguity, ensuring clearer and more straightforward specifications (see App.B.6 for differences). Annotators manually write the instructions, making them natural by *avoiding blunt descriptions*, *removing ambiguity* in the expected results, and ensuring that all SQL *conditions are clearly mentioned*. Also, the DBT-project tasks (see Fig. 1 and App.B.2), which are realistic data transformation coding scenarios, are exclusively used in Spider 2.0. Annotators craft task instructions based on the provided context. After the initial annotation, they verify the semantic equivalence between the SQL queries and instructions, paraphrasing for clarity with the help of LLMs.

**5) Execution-based focused evaluation.** In this step, annotators are required to obtain results from the databases programmatically and write evaluation scripts (details in App.A). The evaluation scripts can process the results in the form of strings, tables, and database files. It is important to note that in table-based evaluations, predicted results may include numerous columns, which might not exactly match the gold standard answers. This discrepancy often arises because some questions do not specify the columns that should be returned. To mitigate this, the evaluation scripts are specifically focused on the essential components of the answers, ignoring non-essential columns and emphasizing the core elements outlined in the instructions. This method facilitates targeted assessments of key columns for each task, thus significantly reducing the occurrence of false negatives. For Spider 2.0-lite and Spider 2.0-snow, these settings require that the output must be SQL, so the evaluation will compare the execution results of the SQLs using the table-based assessment method.

**6) Quality control.** To ensure the quality of our benchmark, each instruction, the gold SQL query, and evaluation script are reviewed by at least three annotators. We require the annotators to repeatedly review steps 3), 4), and 5) to ensure the correctness, naturalness, and unambiguity of the annotations. Consequently, 45% of the examples have errors identified by the first validators. After

discussions and corrections, following the second round of iteration with the second validators, only 5% of the examples contain errors. Then we correct all errors and refine all annotations, and ultimately, all examples are deemed fully annotated. Additionally, we perform a "red team" assessment of our automatic evaluation by providing a set of false results to determine if they would be correctly classified as false, along with various correctly formatted results to verify their classification as true.

## 2.3 DATASET STATISTICS

We present a detailed statistical analysis of the features of Spider 2.0, Spider 2.0-snow and Spider 2.0-lite, comparing them with multiple previous datasets in Tab. 2, our datasets demonstrate strong complexity and realism in aspects such as databases, SQLs, and task scenarios.

Table 2: Statistical comparison among Spider 2.0, Spider 2.0-snow and Spider 2.0-lite, and other text-to-SQL benchmarks. Tok. and Func. refer to tokens and functions, respectively. * denotes the statistics from dev set due to the inaccessibility of test set. For more statistics, refer to App.B.8.

| Dataset | # Test Examples | # Test DB | # Col / DB | # Tok. / SQL | # Func. / SQL | External Knowledge | SQL Dialect | Project Level |
|---|---|---|---|---|---|---|---|---|
| WikiSQL (Zhong et al., 2017) | 15,878 | 5,230 | 6.3 | 12.2 | 0.0 | ✗ | ✗ | ✗ |
| Spider 1.0 (Yu et al., 2018) | 2,147 | 40 | 27.1 | 18.5 | 0.0* | ✗ | ✗ | ✗ |
| KaggleDBQA (Lee et al., 2021) | 272 | 8 | 23.4 | 13.8 | 0.0 | ✓ | ✗ | ✗ |
| SEDE (Hazoom et al., 2021) | 857 | 1 | 212.0 | 46.9 | 1.4 | ✗ | ✗ | ✗ |
| BIRD (Li et al., 2024b) | 1,789 | 15 | 54.2 | 30.9 | 0.4* | ✓ | ✗ | ✗ |
| Spider 2.0-lite | 547 | 158 | 803.6 | 144.5 | 6.5 | ✓ | ✓ | ✗ |
| Spider 2.0-snow | 547 | 152 | **812.1** | **161.8** | 6.8 | ✓ | ✓ | ✗ |
| Spider 2.0 | 632 | 213 | 743.5 | 148.3 | **7.1** | ✓ | ✓ | ✓ |

**Diverse database systems and SQL dialects.** As shown in Fig. 3 and Tab. 3, our benchmarks feature a diverse array of database systems, including cloud data warehouses like *BigQuery* and *Snowflake*, locally hosted databases such as *Postgres* and *ClickHouse*, and lightweight systems like *SQLite* and *DuckDB*. This diversity distinguishes our benchmarks from previous work by encompassing various SQL dialects. Notably, 85.98% of the examples require the use of specialized functions from these dialects, with an average of 7.1 special functions utilized in each ground-truth SQL.

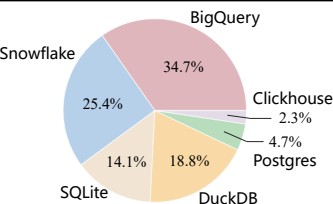

Figure 3: Data distribution on different database systems.

**Real and complex database schema.** As shown in Tab. 2, the databases in Spider 2.0 are equipped with large-scale schemas comprising extensive tables and columns, effectively mirroring real-world enterprise environments. As shown in Tab. 3, these databases are characterized by complex schema structures (e.g., multiple and nested schemas, partitioned tables; see Fig. 11 and Fig. 12), and dynamic tables that are updated daily. Additionally, the data encompasses a broad spectrum of complex types (Fig. 16), extensive volumes, and diverse scopes (Fig. 15), rendering it more diverse than previous datasets.

Table 3: Statistics of Spider 2.0 task features.

| Statistics | Number (% of Total) |
|---|---|
| **Total Levels (#tokens)** | 632 (100%) |
| - Easy   (#tokens < 80) | 160 (25.32%) |
| - Medium   (80 ≤ #tokens < 160) | 279 (44.15%) |
| - Hard   (#tokens ≥ 160) | 193 (30.54%) |
| - With Bigquery | 214 (33.86%) |
| - With Snowflake | 198 (31.33%) |
| - With SQLite | 135 (21.36%) |
| - With DuckDB | 68 (10.76%) |
| - With Postgres | 10 (1.58%) |
| - With Clickhouse | 7 (1.11%) |
| - With Project-level (DBT) | 78 (12.34%) |
| - With Documentation | 82 (12.97%) |
| - With Functions | 474 (75.00%) |
| - With Partition Tables | 54 (8.54%) |
| - With Multiple Schemas | 140 (22.15%) |
| - With Nested Schemas | 117 (18.51%) |
| - With String/Number Answer | 162 (25.63%) |
| - With Table Answer | 392 (62.03%) |
| - With Database Answer | 78 (12.34%) |

**Challenging tasks across the data engineering pipeline.** The examples in our benchmarks are collected from real tutorials and forums, covering a wide range of issues encountered in data pipelines, including data wrangling, data transformation, and data analysis (see App.B.1 for examples). The difficulty of these questions significantly exceeds that of previous SQL-related benchmarks, as the SQL queries in Spider 2.0 contain significantly more columns, tokens, and functions per query than those in prior work (see Tab. 2 and Fig. 18 for examples).

**Real projects scenarios with codebases and documents.** As demonstrated in Tab. 2 and 3, tasks in both datasets require access to documentation, like external knowledge (App.B.4) and SQL dialect (App.B.7), necessitating a deep understanding of these resources. Compared to other prior works,

for each task in Spider 2.0, we provide a codebase context to simulate a real workflow (App.B.5). More notably, some tasks introduce innovations such as project-level data transformation workflows built on DBT (App.B.2), a widely used tool for managing data transformations and analytics engineering. Successfully addressing these tasks requires navigating complex project codebases and databases, comprehending documentation, processing intricate contexts, and generating diverse queries through multi-step execution and reasoning.

## 3 EXPERIMENTS

### 3.1 EXPERIMENTAL SETUP

**Evaluation metrics.** For Spider 2.0, we use the **Success Rate (SR)** metric, which measures the proportion of task instances successfully completed. For Spider 2.0-lite and Spider 2.0-snow, the output for each task must be an SQL, we use the widely used metric **Execution Accuracy (EX)**(Yu et al., 2018; Li et al., 2024b). We employ the execution-based *focused* evaluation (App.A) to determine the success of each result for Spider 2.0 and assess the accuracy of SQL execution results for Spider 2.0-lite. The evaluation scripts are designed to accept output in the form of strings, tables, or database. For each example, an evaluation script is run for each example, producing a score of either 0 or 1. It is worth noting that in table-based evaluations, predicted results may contain numerous columns, leading to results that are not exactly the same as the gold answer. This occurs because, for some examples, questions do not explicitly specify which columns to return. The evaluation scripts are specifically focused on the essential components of the answers, disregarding irrelevant columns and concentrating on the core elements specified in the instructions.

**Difficulty level.** We tokenize the gold SQL queries based on whitespace and classify their difficulty according to the number of tokens: $< 80$ tokens as Easy, $80 - 159$ as Medium, and $\geq 160$ as Hard[1].

**LLMs.** We experiment with state-of-the-art LLMs, including open-source representatives such as DeepseekCoder-V2.5 (Zhu et al., 2024), Qwen2.5-72B-Instruct (Team, 2024) and Llama-3.1-405B (Meta AI, 2024), and closed-source ones including Gemini-Pro-1.5 (Reid et al., 2024), Claude-3.5-Sonnet (Anthropic, 2024) and GPT (OpenAI, 2023) families (GPT-4o, GPT-4, o1-preview and o3-mini). Follow (Yang et al., 2024a; Chen et al., 2024), we use a temperature of 0.0 and truncate from the beginning of the input if still exceeding the max tokens limit required by the models.

**Code agent frameworks.** We utilize several state-of-the-art frameworks, which have demonstrated excellent performance on other benchmarks. These include Reflexion (Shinn et al., 2023), CodeR (Chen et al., 2024), AutoEval (Pan et al., 2024). Inspired by React (Yao et al., 2022) and Intercode (Yang et al., 2023), we develop an agent framework called Spider-Agent, which is primarily focused on database-related coding tasks and projects. The framework allows for multi-turn interactions with the database via command-line interfaces until the final answer is obtained. The implementation details of Spider-Agent are shown in App.C.1.

**Text-to-SQL methods.** We also evaluate several state-of-the-art and widely recognized text-to-SQL methods, including approaches based on prompting LLMs such as DIN-SQL (Pourreza & Rafiei, 2024), DAIL-SQL (Gao et al., 2024) and CHESS (Talaei et al., 2024), alongside SFT CodeS (Li et al., 2024a), which fine-tuned open-source models on extensive text-to-SQL corpora. DAIL-SQL and CHESS achieve the best performance among all accessible methods on the Spider 1.0 and BIRD benchmark, respectively. During implementation, we optimize the prompt organizations across all methods to better align with tasks, incorporating sampled cell values, external knowledge, and SQL dialect specifications (see Fig. 21).

### 3.2 EVALUATION RESULTS

**Existing LLMs are still far from being expert on real-world text-to-SQL workflow tasks.** As shown in Tab.4 and Tab.6, we used Spider-Agent and its variants to conduct tests on Spider 2.0, Spider 2.0-Lite, and Spider 2.0-Snow. The o1-preview and o3-mini achieve the highest performance, with a peak success rate of 23.77% on Spider 2.0-snow and 23.40% on Spider 2.0-lite, indicating significant potential for further improvement. It surpasses both GPT-4o and Claude-3.5-Sonnet across the *Easy*, *Medium*, and *Hard* cases, highlighting its superior reasoning capabilities. The open-source LLM DeepSeek-V3 showed a performance of 8.78%, still has significant room for improvement. The results shown in Tab.6, combined with the DBT project examples, also exhibit a similar

---

[1]While there are various ways to measure difficulty, we use SQL length here as the most common and significant metric for experimental reference.

Table 4: Execution Accuracy (EX) of different models using Spider-Agent on Spider 2.0-lite and Spider 2.0-snow, grouped by difficulty level.

| Model | Spider 2.0-Lite | | | | Spider 2.0-Snow | | | |
|---|---|---|---|---|---|---|---|---|
| | Easy | Medium | Hard | **Overall** | Easy | Medium | Hard | **Overall** |
| o1-preview | 33.59% | 23.58% | 15.03% | 23.22% | 39.84% | 21.14% | 15.61% | **23.77%** |
| o3-mini | 32.03% | 26.02% | 13.87% | **23.40%** | 31.25% | 18.29% | 11.56% | 19.20% |
| Claude-3.5-Sonnet | 26.56% | 15.85% | 6.94% | 15.54% | 25.00% | 16.26% | 7.51% | 15.54% |
| GPT-4o | 22.66% | 13.41% | 5.78% | 13.16% | 24.22% | 11.38% | 6.94% | 12.98% |
| DeepSeek-V3 | 19.53% | 6.50% | 4.05% | 8.78% | 20.31% | 6.1% | 4.05% | 8.78% |
| Qwen2.5-Coder | 13.89% | 4.17% | 3.38% | 5.30% | 11.72% | 4.47% | 2.31% | 5.48% |

Table 5: Execution Accuracy (EX) for baseline methods on three text-to-SQL datasets: Spider 1.0, BIRD, Spider 2.0-lite and Spider 2.0-snow.

| Method | EX (↑) | | | | | | |
|---|---|---|---|---|---|---|---|
| | Spider 1.0 | BIRD | Spider 2.0-snow | Spider 2.0-lite | | | |
| | | | | Easy | Medium | Hard | **Overall** |
| DIN-SQL + GPT-4o | 85.3% | 55.9% | 0.00% | 5.79% | 0.43% | 0.00% | 1.46% |
| DAIL-SQL + GPT-4o | 86.6% | 57.4% | **2.20%** | **13.20%** | **5.58%** | 1.24% | **5.68%** |
| CHESS + GPT-4o | 87.2% | 66.7% | 1.28% | 9.92% | 3.00% | **1.24%** | 3.84% |
| SFT CodeS-15B | 85.4% | 59.3% | 0.00% | 1.65% | 0.86% | 0.00% | 0.73% |

trend. Tab. 5 illustrates that Spider 2.0-lite and Spider 2.0-snow present significant challenges for traditional text-to-SQL methods. The highest performing method, DAIL-SQL + GPT-4o, achieves an EX of only 5.68%, which is markedly lower compared to its score of 86.6% on Spider 1.0 and 57.4% on BIRD datasets. With efficiently filtering the minimal sufficient schema, CHESS + GPT-4o is able to tackle more instances than DIN-SQL. Despite being extensively fine-tuned, SFT CodeS-15B is far from solving Spider 2.0-lite, with an EX score of only 0.73%, which further reveals the significant complexity gap between Spider 2.0-lite and the current text-to-SQL corpus. For Spider 2.0-snow, even the best method achieves only 2.20% EX, highlighting the increased challenge due to SQL dialect differences.

**Existing code agent frameworks struggle with solving database-related coding tasks.** Tab.4 and Tab. 6 show that the current agent frameworks are still unable to effectively address the tasks. The challenge is that they must not only explore the codebase and documentation, but also navigate complex databases and generate SQL queries that are far more intricate than typical code. This demands a high level of code grounding capability. Spider-Agent provides a crucial baseline for Spider 2.0, facilitating the evaluation of various LLMs, underscoring the potential for significant advancements and inspiring methodology enhancements for future research. We also observe that the model must be proficient in debugging from SQL execution feedback and exploring the schemas of different types of databases (e.g., Snowflake), which poses a significant challenge to the code agent's capabilities. There is still significant room for improvement in Spider-Agent when it comes to enterprise-level SQL tasks, in order to fully unleash LLMs' text-to-SQL capabilities.

Table 6: Success rate (SR) of different frameworks and models on Spider 2.0. The costs under different settings are shown in Tab.21. Spider 2.0 consists of Spider 2.0-Lite along with DBT-project tasks.

| Framework | Model | SR (↑) |
|---|---|---|
| AutoEval | GPT-4o | 5.70% |
| Reflexion | GPT-4o | 7.28% |
| CodeR | GPT-4o | 7.91% |
| Spider-Agent | o1-Preview | **21.36%** |
| | Claude-3.5-Sonnet | 14.87% |
| | GPT-4o | 12.34% |
| | GPT-4 | 9.86% |
| | Qwen2.5-72B | 6.17% |
| | DeepSeek-V2.5 | 5.22% |
| | Gemini-Pro-1.5 | 2.53% |
| | Llama-3.1-405B | 2.21% |

## 4 ANALYSIS

### 4.1 ANALYSIS OF DIFFERENT TASK TYPES

**LLM-agent frameworks struggle interpreting databases with nested schema.**

As shown in Tab. 7, the model often performs poorly when handling columns with nested types. Nested columns are a common scenario in industrial-grade databases (see Fig. 11), where data is stored in array, dict formats within a single column. This poses significant challenges for LLMs in understanding the schema. As shown in Fig. 29, LLMs encounter schema linking errors due to an incomplete understanding of the information contained within nested fields. Most databases with nested types face the issue that models find it difficult to fully grasp the function of each nested column's internal information, while humans can comprehend the schema through multi-step reasoning and iterative understanding.

Table 7: Model performance on databases with nested columns in non-dbt projects.

| Task Subset | % of Total | SR (↑) |
|---|---|---|
| w/ Nested Column | 18.51% | 10.34% |
| w/o Nested Columns | 68.04% | 27.38% |

**The performance drops when external documents are required.**

From Tab. 8, we observe that when tasks involve external documents, the model performs poorly, correctly answering only 11 examples out of full dataset that accounts for just 11.54%. Through error analysis, we find that the model is not incapable of grounding complex documents information. These models typically have the correct problem-solving strategies and effectively explore the database, but fails at the most crucial step: grounding the complex requirements from the documents into SQLs. As the document shown in Fig. 13, the gold SQL is shown in Tab. 16. The failure case shows that the model cannot combine complex document with schema information and convert it into SQL query (Fig. 28).

Table 8: Performance of the model on external document tasks in non-dbt projects.

| Task Subset | % of Total | SR (↑) |
|---|---|---|
| w/ External Doc | 12.97% | 11.54% |
| w/o External Doc | 73.58% | 26.64% |

**LLM-agent frameworks struggle to address project-level tasks.**

As shown in Tab. 9, the LM agent's performance on DBT-based project tasks is poor, solving only 12.82% of tasks with just 10 examples correct. This underscores the challenges in there tasks, which can be attributed to: (1) Data transformation projects often require *multiple SQL queries* to complete various models, necessitating a comprehensive understanding of the project. (2) These tasks involve *complex context usage*, demanding strong repository exploratory capabilities. (3) Data is stored in databases, requiring the agent to transform data while exploring existing data, alongside SQL coding. Fig. 26 illustrates the action process of o1-preview successfully solving a task defined in App.B.2, while Fig. 27 is a failure case due to the failure to explore the information in the "mrr.md" file to solve a monthly recurring revenue classification.

Table 9: Performance on DBT Project.

| Task Subset | % of Total | SR (↑) |
|---|---|---|
| w/ DBT Project | 12.34% | 12.82% |
| w/o DBT Project | 87.65% | 23.22% |

### 4.2 ERROR ANALYSIS OF SQL GENERATION

We conduct a detailed analysis of the errors encountered by both code agent frameworks on randomly sampled 300 examples, as illustrated in Fig. 4. Representative errors along with their statistics and causal analysis are as follows.

**Erroneous data analysis (35.5%).** Compare to the previous benchmarks, Spider 2.0 and Spider 2.0-lite exhibit significantly complex data analysis demands that challenge the models' capabilities: 1) Dialect function usage (10.3%). This includes processing temporal (e.g., DATE_TRUNC) or geographic data (e.g., ST_DISTANCE). These functions require a nuanced understanding, which the models often fail to exhibit. 2) Advanced data calculation (7.5%). Model struggle with tasks like grouping samples to analyze trends within groups (using NTILE), or applying formulas for statistical values (e.g., CORR for Pearson correlation coefficients; STDDEV for standard deviation). 3) Intricate query planning (17.7%). Gold SQLs typically involve multiple nested queries, in-

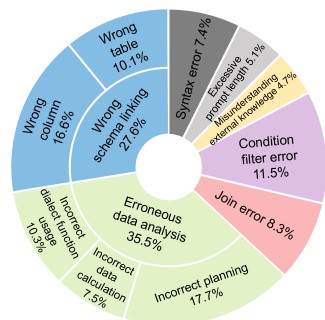

Figure 4: Statistics of errors. For detailed descriptions and examples of each error category, see App.C.3.

termediate result processing through common table expressions (CTEs), or merging results from various sub-queries via set operations. However, models often inadequately handle these complexities. Refer to Fig. 5 for case studies on erroneous data processing.

**Wrong schema linking (27.6%).** This category includes errors with wrong tables and columns. For column linking errors (16.6%), the average number of columns per database in Spider 2.0-lite far exceeds those in other benchmarks (over 755 compared to approximately 54 in BIRD), making accurate column linking extremely challenging. Regarding table linking (10.1%), although examples from BigQuery support advanced syntax features like (TABLE_SUFFIX) and wildcard expressions, the models show limited flexibility in leveraging these features, even in few-shot setting.

**JOIN errors (8.3%).** While foreign keys represent known schema relationships essential for valid SQL JOIN operations, databases in BigQuery often lack explicit foreign key. This omission forces models to infer potential keys based on column names and descriptions, leading to errors.

Table 10: EX for baseline methods on Spider 2.0-lite under oracle setting. To seek the highest possible performance, we also employ the latest o1-preview as the base LLM.

| Method | EX (↑) | |
|---|---|---|
| | w.Oracle Func | w/o Oracle Func |
| DAIL-SQL + GPT-4o | 5.85% | 5.68% |
| DAIL-SQL + o1-preview | 9.51% | **12.60%** |

Table 11: EX for DAIL-SQL on Spider 2.0-lite under few-shot setting with manually selected demonstrations.

| Method | EX (↑) | | |
|---|---|---|---|
| | 0-shot | 1-shot | 3-shot |
| DAIL-SQL + GPT-4o | 5.68% | 6.40% | 6.76% |

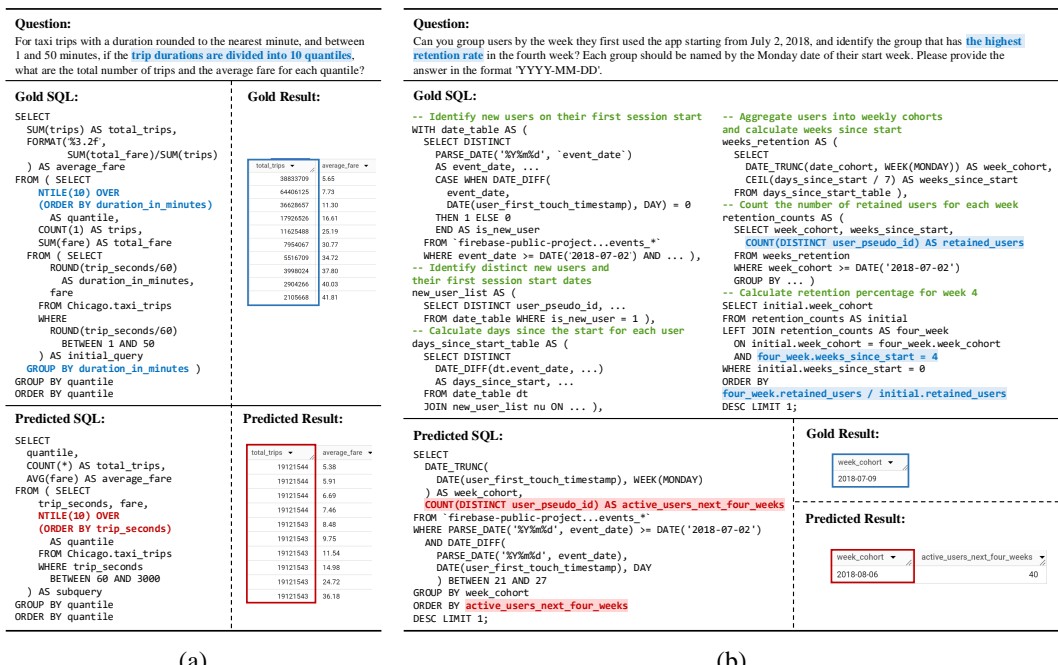

(a)                    (b)

Figure 5: Case study of two representative incorrect SQL predictions due to erroneous data analysis. (a): An example of **incorrect data calculation**, where quantiles were incorrectly divided based on the number of trips, rather than on the *trip duration* as required. (b): An example of **incorrect planning**, where the predicted SQL incorrectly sorted data by the number of users, rather than by the required *retention ratio*. The prerequisite for achieving this is to properly plan a sequence of CTEs. Additional examples of error cases across all categories are available in Fig. 22 and Fig. 23.

## 4.3 ANALYSIS OF DIFFERENT EXPERIMENTAL SETTINGS

**Providing oracle functions leads to a slight performance improvement.** Considering that Spider 2.0 and Spider 2.0-lite involve SQL dialects from various database systems, we provide syntax and

function documentation for each system to prevent the methods from suffering due to lack of syntax knowledge. For each example, we manually include the relevant function documentation that may be required, eliminating the need for a retrieval method and ensuring that the necessary syntax knowledge is readily accessible. As shown in Tab. 10, providing oracle SQL function documentation results in only a slight improvement in model performance. This suggests that, to a certain extent, models are capable of selecting appropriate functions and understanding their basic usage and syntax. However, the critical challenge lies in accurately utilizing these functions to reflect user intentions, as illustrated in Fig. 5(a).

**Few-shot prompting has little impact on performance.** Spider 2.0-lite is not divided into train and dev sets, we manually select representative examples from the same SQL dialect as the SQL to be predicted, with distinct characteristics (encompassing multiple CTE or nested queries, or requiring intricate data processing) to serve as few-shot examples. Unexpectedly, few-shot in-context learning shows only marginal improvements in performance (see Tab. 11). This may be due to the gap between the simplistic text-to-SQL pre-training data used with LLMs and the complexity of the few-shot examples. Additionally, extensive schema prompts may hinder the model's ability to effectively assimilate information in the few-shot examples.

## 5   RELATED WORK

**Code generation and text-to-SQL benchmark.** As model capabilities advance, code generation benchmarks have become more complex and generalized. Many benchmarks (e.g., SQL-Spider (Yu et al., 2018), Bash-NL2Bash (Lin et al., 2018), Python-HumanEval (Chen et al., 2021)) treat code generation as seq2seq tasks. Many previous works (Lai et al., 2023; Yin et al., 2023; Huang et al., 2024; Chan et al., 2024; Jing et al., 2024) define code generation tasks for data science. MLAgent-Bench (Huang et al., 2023) and Intercode (Yang et al., 2024b) focus on interactive environments, while SWE-Bench (Jimenez et al., 2023) emphasizes repository-level coding tasks. Spider2-V (Cao et al., 2024) proposes data science and engineering benchmark in a multimodal setting. Many previous datasets (Zhong et al., 2017; Lee et al., 2021; Hazoom et al., 2021; Wang et al., 2020; Li et al., 2024b) have made significant contributions to the advancement of text-to-SQL tasks. However, existing text-to-SQL benchmarks primarily target lightweight local databases, much smaller in schema scale and data volume than cluster-hosted industrial databases, and fail to capture the *agentic* nature of SQL programming using various dialects in real scenarios. Spider 2.0 bridges the gap between research and enterprise-level industrial text-to-SQL workflows.

**Code agent framework and text-to-SQL methods.** The intersection of generative code models and interactive problem-solving has spurred significant advancements in both agent-based frameworks and text-to-SQL methodologies. Recent efforts aim to enhance the reasoning capabilities of language models, as evidenced by a surge in agent methods designed for code generation tasks (Yao et al., 2022; Zhang et al., 2022; Chen et al., 2023; Wang et al., 2023b; Shinn et al., 2024; Zhang et al., 2024; Xia et al., 2024). Several works have designed special actions to standardize agent operations (Wang et al., 2024; Yang et al., 2024a). For methods specifically designed for text-to-SQL, several fine-tuning methods (Li et al., 2024a) and LLM-prompting methods (Dong et al., 2023; Wang et al., 2023a; Zhang et al., 2023; Talaei et al., 2024; Pourreza & Rafiei, 2024; Gao et al., 2024) have achieved strong performance on previous benchmarks. We propose Spider-Agent, a code agent framework specifically designed for database-related tasks, showcasing strong performance in this domain. For Spider 2.0-lite, we also adapt several text-to-SQL methods to suit our benchmark.

## 6   CONCLUSION

We propose Spider 2.0, a benchmark for real-world enterprise-level text-to-SQL workflow tasks. It encompasses diverse database systems with various SQL dialects, large and complex database schemas, and challenging tasks across the data engineering pipeline, all set within real project scenarios including codebases and documentation. Despite being the most advanced LLMs (o1-preview), they still perform poorly on Spider 2.0, achieving a success rate of only 21.3%, which underscores its status as a highly challenging benchmark. Spider 2.0 presents a novel challenge for text-to-SQL research, providing a direction towards more realistic and intelligent solutions.

ACKNOWLEDGEMENTS

The authors of this paper were supported by the ECS (27212023) from RGC of Hong Kong. We thank Snowflake for their generous support in hosting the Spider 2.0 Challenge. We also thank Tianbao Xie, Yiheng Xu, Fan Zhou, Yuting Lan, Per Jacobsson, Yiming Huang, Canwen Xu, Zhewei Yao and Binyuan Hui for their helpful feedback on this work.

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

## A SPIDER 2.0 EVALUATION SCRIPTS

In this section, we present the detailed definition and discussion of evaluation metrics for Spider 2.0-lite and Spider 2.0.

**Spider 2.0-lite.** The setting of Spider 2.0-lite resembles that of a traditional text-to-SQL task in which text-to-SQL parsers are required to generate SQL queries. Therefore, **Execution Accuracy(EX)** is used as the primary evaluation metric. Slightly different from existing works, we employ an *execution-based focused evaluation*, which measures whether all columns in the gold value are present in the output of the predicted SQL query. This is defined as follows:

$$\text{EX} = \frac{\sum_{n=1}^{N} \mathbb{1}(v^n, \hat{v}^n)}{N}, \tag{1}$$

$$\text{where} \quad \mathbb{1}(v, \hat{v}) = \begin{cases} 1, \text{ if } v_i \in \hat{v}, \forall v_i \in v \\ 0, \text{ if } v_i \notin \hat{v}, \exists v_i \in v \end{cases}, \tag{2}$$

where $v_i$ represents the $i$-th column of data frame $v$, $v^n$ and $\hat{v}^n$ denote execution results of the gold SQL and predicted SQL for the $n$-th instance in the evaluation set, respectively. Empirically, this evaluation method significantly reduces the false negative rate without increasing the number of false positives. Given that the ground-truth values result from extensive data wrangling, transformation, and analysis, it is difficult for models to manipulate or exploit the system.

**Spider 2.0.** We use the **Success rate (SR)**, which measures the proportion of task instances successfully resolved. Human-written evaluation scripts are used to determine whether an example is resolved. For each example, we provide string-based, table-based, and database-based evaluation functions, depending on the type of answer output, as shown in Tab. 12.

**Examples.** Maintaining naturalness and unambiguity is often a conflicting challenge. To address this, we provide an example to illustrate the important parameters "condition_cols" and "ignore_order". Achieving a balance between these two aspects is quite challenging, which is why we incorporate this mechanism into our evaluation scripts.

Given a data frame $v$ with a set of column vectors $\{v_i\}$, each representing the cell values for the $i$-th column, a prediction $\hat{v}$ is considered equivalent with $v$ if and only if for any $v_i \in v$, $v_i \in \hat{v}$. Therefore, at such times, we only check whether a specific column appears in it. Intuitively, if all columns in the reference table appear in the result table, the result is considered correct.

For example, as illustrated in Fig. 6, the question does not explicitly specify which columns are required in our response. Consider the following question: *"The company management has requested a detailed report on the year-to-date performance of the Magnificent 7 stocks."*. We need to carefully analyze the task requirements and only check if the following columns in the reference answer—*"Ticker"*, *"Change_YTD"*—appear in the predicted answer. This meets the semantic requirements of the abstract instruction. Empirically, we find our evaluation metric is reliable in identifying solutions with alternative output, with a relatively low false-negative rate.

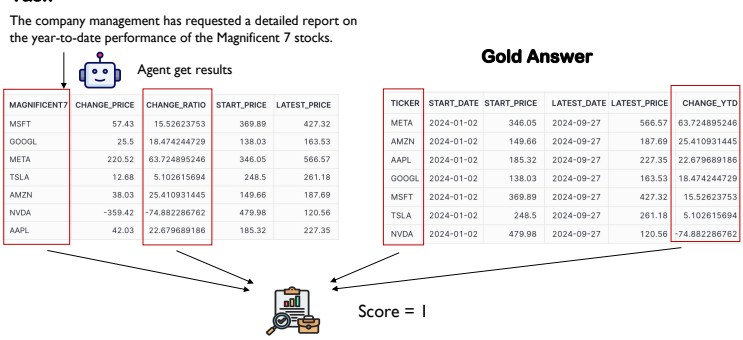

Figure 6: An example of evaluation scripts for table-based evaluation: in this example, the `condition_cols` is $\{0, 5\}$, and the `ignore_order` is *true*. As long as these two columns are predicted correctly, the example can be considered solved.

Table 12: The evaluation scripts for Spider 2.0 are tailored to the specific format of the model's output. Each script is optimized to handle various output types, ensuring precise and contextually appropriate evaluation.

| Output Type | Description | Parameters |
|---|---|---|
| String w/o number | *If the answer is found in the string, it is given a score of 1; otherwise, it receives a score of 0.* | **pred** (str): The string in which to search for substrings. **gold** (List of str): A list of strings to check within the predicted string. **conj** (str): The conjunction used for matching ('and' or 'or'). Default is 'or'. **exclude** (List of Str): Strings that must not be present in the answer. |
| String w. number | *For output strings containing numbers, the script captures these numbers and performs number matching for scoring using the number_match function.* | **pred** (str): The string in which to search for substrings. **gold** (List[str\|float]): A list of strings or numbers to check within the predicted string. **percentage** (bool): Default is false. If the gold answer is related to percentages, set this to true for more robust evaluation. **precision** (int): The number of decimal places to consider. Defaults to 4. **conj** (str): The conjunction used for matching ('and' or 'or'). Default is 'or', and it's typically 'or'. |
| Table | *If the answer is a CSV file or a table in string format, table-level evaluation is performed.* | **result** (str): Path to the CSV file or result string. **gold** (str \| List[str]): Path(s) to the gold file(s), excluding the root directory. Multiple potential gold answers are supported. **condition_cols** (List[int] \| List[List[int]]): List of column indices to match conditions. For example, [0, 1] uses the 0th and 1st columns in the gold table for matching, while ignoring the others. **ignore_order** (bool): Whether to ignore the order of rows when matching elements. |
| Database | *If the answer is stored in a DB file, database-level evaluation is applied using the db_match function.* | **result** (str): Path to the DuckDB file containing the result tables. **gold** (str): Path to the DuckDB file containing the gold standard tables. **condition_tabs** (List[str], optional): List of table names to be checked. If not provided, all tables in the gold DuckDB file will be considered. **condition_cols** (List[List[int]], optional): A list of lists, where each inner list contains column indices used for matching conditions for the corresponding table. Defaults to considering all columns. **ignore_orders** (List[bool], optional): A list of boolean values indicating whether to ignore the row order for each table comparison. Defaults to [False] for each table. |
| SQL | *If the output is an SQL, execution-based evaluation is used. This is primarily designed for Spider 2.0-lite.* | To compare the execution results of the predicted SQL and the gold SQL, table matching is used. |

# B   ANNOTATION DETAILS

## B.1   SQL ANNOTATION EXAMPLES

In this section, we present several representative examples of the SQL annotation process, including the original SQL, how the SQL was rewritten to obtain the gold SQL, and the use of external knowledge.

Tab. 13 presents an example based on the Google Analytics database. The task is to calculate the source of web traffic and count the number of sessions for each traffic channel within a given time period.

Tab. 14 presents an example based on New York City public data, where the task is to find Citibike and taxi trips between specified locations and analyze whether Citibike or taxi is more suitable for travel between the two locations. In this case, the condition in the original SQL is to calculate trips between the two locations by Citibike and car. We extend this condition by introducing a real-life problem: identifying which routes are faster by Citibike compared to taxi.

Tab. 15 is based on the Google Patents database, which contains a large amount of patent information. The original SQL applied several filtering conditions to retrieve a set of patents. We find a document explaining how to calculate a patent's originality score, which led to an advanced calculation method. As a result, the final task include additional complex calculation steps.

Tab. 16 is also based on the Google Analytics database. The original SQL calculates the Product List Page (PLP) and Product Details Page (PDP). Based on the description in the blog, we define a new task to calculate the conversion rate by determining the probability of users clicking from PLP to PDP.

Tab. 17 presents an example where we merge and rewrite two related SQL queries. The first SQL calculates the 50 weather stations closest to downtown Chicago, while the second SQL calculates the number of bike trips on rainy and non-rainy days in New York City. We combine these two tasks, meaning that to determine whether it is a rainy day, we first need to find data from the weather station closest to downtown New York City.

## B.2   DBT PROJECT ANNOTATION EXAMPLES

**Annotation Pipeline of DBT Project.** The DBT project can be found on online resources and is one of the projects with the most SQL scripts. Similar data transformation tools are widely used in industrial production. Completing a DBT project requires a comprehensive understanding of both the code and documentation within the project to accomplish the entire task. Fig. 7 shows a Salesforce-based project in Spider 2.0. This represents a natural and realistic SQL generation scenario. Using a Fivetran Salesforce transformation package [2] as an example, we transform a complex DBT project into a Spider 2.0 example through the following steps.

(1) Run a DBT project from start to finish, ensuring it is bug-free and generates a dbt DAG (Fig. 9). This allows for a comprehensive understanding of the data flow.

(2) The DBT project includes yml files and markdown documents, where the project developers have already planned out the data models and data flow. We will use these as the basis for task instructions.

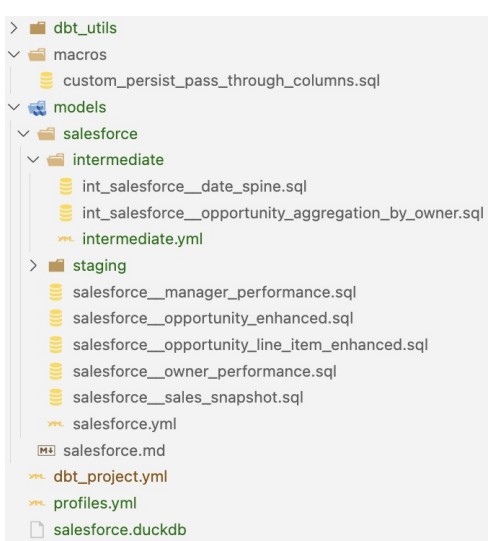

Figure 7: Codebase for a DBT project, showing models, macros, and configuration files.

---

[2] https://github.com/fivetran/dbt_salesforce/

```
- name: salesforce__opportunity_enhanced
  description: >
    This table both cleans and enhances your opportunities source table with information about
    the associate account and opportunity owner.
  ref:
      - name: opportunity
      - name: user
      - name: user_role
      - name: account
  columns:
    - name: opportunity_id
      description: The unique, system-generated ID assigned during creation.
      tests:
        - not_null
        - unique
    - name: account_number
      description: Number of the account associated with this opportunity.
    - name: account_id
      description: ID of the account associated with this opportunity.
    - name: account_type
      description: Type of account, for example, Customer, Competitor, or Partner.
    - name: amount
      description: Estimated total sale amount. For opportunities with products, the amount is the sum of the related products.
    - name: campaign_id
      description: ID of a related Campaign. This field is defined only for those organizations that have the campaign feature Campaigns enabled.
    - name: close_date
      description: Required. Date when the opportunity is expected to close.
    - name: created_date
      description: Date when the opportunity is was created.
    - name: opportunity_description
      description: Text description of the opportunity.
    - name: expected_revenue
      description: Read-only field that is equal to the product of the opportunity Amount field and the Probability.
    - name: fiscal
      description: If fiscal years are not enabled, the name of the fiscal quarter or period in which the opportunity CloseDate falls.
    - name: fiscal_quarter
      description: Represents the fiscal quarter. Valid values are 1, 2, 3, or 4.
    - name: fiscal_year
      description: Represents the fiscal year, for example, 2006.
    - name: forecast_category
      description: The category of the state of the opportunity. Default values are 'pipeline','forecast','bestcase','closed', and 'omited'
    - name: forecast_category_name
      description: The name of the forecast category.
    - name: has_open_activity
      description: Indicates whether an opportunity has an open event or task (true) or not (false).
    - name: has_opportunity_line_item
      description: Read-only field that indicates whether the opportunity has associated line items. A value of true means that Opportunity line i
    - name: has_overdue_task
      description: Indicates whether an opportunity has an overdue task (true) or not (false).
    - name: last_activity_date
      description: Value is one of the following, whichever is the most recent:Due date of the most recent event logged against the record or Due
    - name: last_referenced_date
      description: The timestamp when the current user last accessed this record, a record related to this record, or a list view.
    - name: last_viewed_date
      description: The timestamp when the current user last viewed this record or list view. If this value is null, the user might have only acces
    - name: lead_source
      description: Source of this opportunity, such as Advertisement or Trade Show.
    - name: opportunity_name
```

Figure 8: This is a common configuration file in DBT projects used to define the schema of a data model. It represents a natural SQL generation scenario, specifying details such as field names, data types, and references for the "salesforce_opportunity_enhanced" data model.

(3) We remove the .sql files corresponding to a specific data flow within the complete DBT project. For example, in Fig. 9, we may delete one to three data flows, as shown in Fig. 10, removing "sales_daily_activity" and "salesforce_contact_enhanced" along with their upstream nodes. This turns it into an incomplete transformation project. Note that the DAG figure is only used as an aid for data annotation, and the task does not include any images.

(4) Write the task instructions. For instance, we can draft a prompt like, "I need a daily report on key sales activities—covering tasks completed, events held, leads generated, and the status of opportunities." Although the data model contains many columns, thanks to the presence of yml files (see Fig. 8), there is no need to describe the output columns in detail in the instructions.

**Approach to Solving DBT Project Examples.** As shown in Fig. 26, completing a DBT project example typically requires the following abilities:

1) Problem comprehension. First, it is necessary to fully understand a natural language task.

2) Project reading ability. A real-world data transformation project consists of multiple files, as illustrated in Fig. 7. The method needs to explore the codebase and review relevant project files, including *.yml*, *.md*, and *.sql* files. YML files (Fig. 8) generally define the data models for the data transformation, *.md* files contain textual descriptions of the data models, and SQL files are the data transformation models themselves.

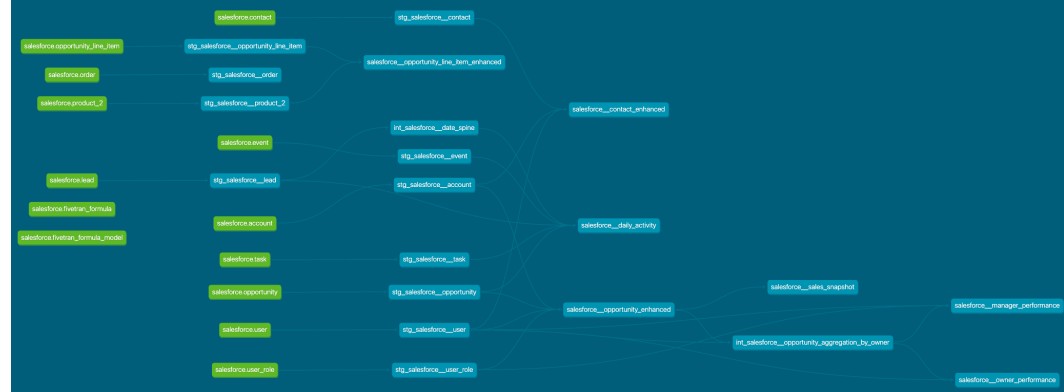

Figure 9: A DAG (Directed Acyclic Graph) illustrating the data flow and dependencies between various Salesforce tables and models in a dbt (data build tool) project. The graph shows stages of transformation, from raw Salesforce data (green nodes) to enhanced and aggregated models (blue nodes), representing different entities such as opportunities, contacts, accounts, and events.

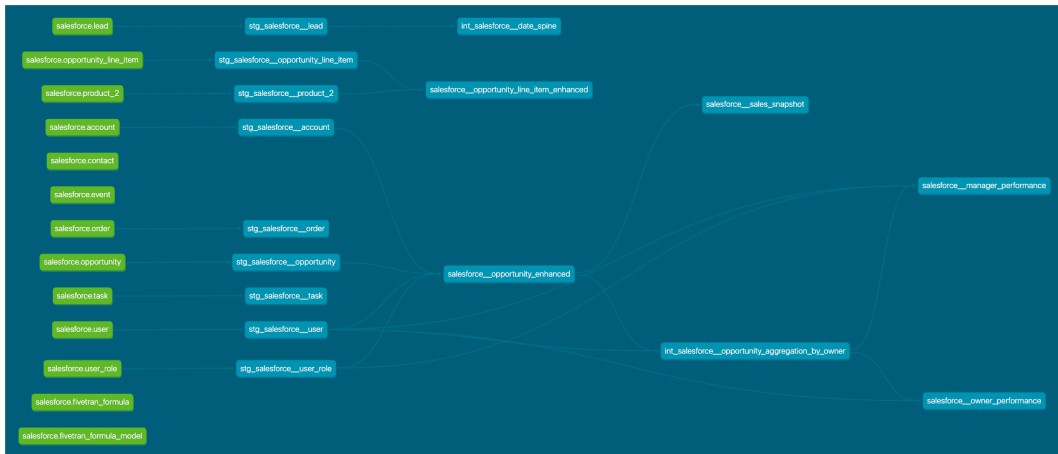

Figure 10: In this version of the DAG, several data models are missing, including "salesforce_daily_activity" and "salesforce_contact_enhanced" along with their upstream nodes. This creates an incomplete data flow compared to the original.

3) Database exploration ability. The codebase only contains the data transformation code, while the data to be transformed is stored in a database. The method must explore the database to understand the available source data and identify any missing data models.

4) Problem localization ability. By combining the natural language problem and the YML files, the method should locate where to add or modify the code in the project.

5) Coding ability. The method needs to complete complex data transformation code based on the data models defined in the YML files and add the .sql files in the appropriate locations. Visually, it requires completing the data models defined in the yml file, transitioning from Fig. 10 to Fig. 9.

6) Data transformation execution. Once the SQL is written, it is necessary to run `dbt run` to execute the data transformation.

7) Debugging. After running the DBT project, if the data transformation is successful, the data models (the tables) in the database will change, with tables being added or removed. The method needs to examine the database to determine if the transformation was fully successful. If not, the above steps must be repeated until the method meets the problem's requirements.

Table 13: Google analytics traffic session examples, using *answer format surface* rewrite.

---

**Question**

---

Provide the number of sessions and percentage breakdown by channel for December 2020.

---

**Reference Plan**

---

1. First, read the document to understand how traffic is divided into 18 channel groups, primarily based on the metrics of source, medium, and campaign. 2. Extract all visits from the database for December, each visit having a unique user ID and session ID. Retrieve the source, medium, and campaign for each visit. 3. Based on the classification standards for channel groups in the document, write conditional statements to determine which channel each set of data belongs to, mainly using regular expressions. If the data source (source) contains any of the following: 'badoo', 'facebook', 'fb', 'instagram', 'linkedin', 'pinterest', 'tiktok', 'twitter', or 'whatsapp', and the medium (medium) includes 'cp', 'ppc', or starts with 'paid', then categorize it as 'Paid Social'. 4. Calculate the number of sessions and percentage for each channel based on the channel grouping.

---

**Gold SQL (After rewriting)**

```sql
WITH prep AS (
SELECT user_pseudo_id,
 (SELECT value.int_value FROM UNNEST(event_params) WHERE key = 'ga_session_id') AS session_id,
       ARRAY_AGG((SELECT value.string_value FROM UNNEST(event_params) WHERE key = 'source')
       IGNORE NULLS ORDER BY event_timestamp)[SAFE_OFFSET(0)] AS source,
       ARRAY_AGG((SELECT value.string_value FROM UNNEST(event_params) WHERE key = 'medium')
       IGNORE NULLS ORDER BY event_timestamp)[SAFE_OFFSET(0)] AS medium,
       ARRAY_AGG((SELECT value.string_value FROM UNNEST(event_params) WHERE key = 'campaign')
       IGNORE NULLS ORDER BY event_timestamp)[SAFE_OFFSET(0)] AS campaign
FROM `bigquery-public-data.ga4_obfuscated_sample_ecommerce.events_*`
WHERE _TABLE_SUFFIX BETWEEN '20201201' AND '20201231' GROUP BY user_pseudo_id, session_id )
grouped_data AS (
SELECT CASE
WHEN source = '(direct)' AND (medium IN ('(not set)', '(none)')) THEN 'Direct',
     WHEN REGEXP_CONTAINS(campaign, 'cross-network') THEN 'Cross-network'
WHEN (REGEXP_CONTAINS(source, 'alibaba|amazon|google shopping|shopify|etsy|ebay|stripe|walmart')
     OR REGEXP_CONTAINS(campaign, '^(.*(([^a-df-z]|^)shop|shopping).*)$'))
     AND REGEXP_CONTAINS(medium, '^(.*cp.*|ppc|paid.*)$') THEN 'Paid Shopping' ,
WHEN REGEXP_CONTAINS(source, 'baidu|bing|duckduckgo|ecosia|google|yahoo|yandex')
                     AND REGEXP_CONTAINS(medium, '^(.*cp.*|ppc|paid.*)$') THEN 'Paid Search' ,
WHEN REGEXP_CONTAINS(source, 'badoo|facebook|fb|instagram|linkedin|pinterest|tiktok|twitter|whatsapp')
                     AND REGEXP_CONTAINS(medium, '^(.*cp.*|ppc|paid.*)$') THEN 'Paid Social' ,
WHEN REGEXP_CONTAINS(source, 'dailymotion|disneyplus|netflix|youtube|vimeo|twitch|vimeo|youtube')
                     AND REGEXP_CONTAINS(medium, '^(.*cp.*|ppc|paid.*)$') THEN 'Paid Video' ,
WHEN medium IN ('display', 'banner', 'expandable', 'interstitial', 'cpm') THEN 'Display'
WHEN REGEXP_CONTAINS(source, 'alibaba|amazon|google shopping|shopify|etsy|ebay|stripe|walmart')
     OR REGEXP_CONTAINS(campaign, '^(.*(([^a-df-z]|^)shop|shopping).*)$') THEN 'Organic Shopping'
WHEN REGEXP_CONTAINS(source, 'badoo|facebook|fb|instagram|linkedin|pinterest|tiktok|twitter|whatsapp')
     OR medium IN ('social', 'social-network', 'social-media', 'sm', 'social network', 'social media') THEN 'Organic Social'
WHEN REGEXP_CONTAINS(source, 'dailymotion|disneyplus|netflix|youtube|vimeo|twitch|vimeo|youtube')
                     OR REGEXP_CONTAINS(medium, '^(.*video.*)$') THEN 'Organic Video'
WHEN REGEXP_CONTAINS(source, 'baidu|bing|duckduckgo|ecosia|google|yahoo|yandex') OR medium = 'organic'  THEN 'Organic Search'
WHEN REGEXP_CONTAINS(source, 'email|e-mail|e_mail|e mail') OR REGEXP_CONTAINS(medium, 'email|e-mail|e_mail|e mail') THEN 'Email'
WHEN medium = 'affiliate' THEN 'Affiliates' WHEN medium = 'referral' THEN 'Referral'
WHEN medium = 'audio' THEN 'Audio' WHEN medium = 'sms' THEN 'SMS'
WHEN medium LIKE '%push' OR REGEXP_CONTAINS(medium, 'mobile|notification') THEN 'Mobile Push Notifications' ELSE 'Unassigned'
END AS channel_grouping_session, COUNT(DISTINCT CONCAT(user_pseudo_id, session_id)) AS session_count
FROM
 prep GROUP BY channel_grouping_session
),
total_sessions AS (
SELECT SUM(session_count) AS total_count FROM grouped_data )
SELECT  gd.channel_grouping_session, gd.session_count, ROUND(gd.session_count / ts.total_count * 100, 2) AS percentage_of_total
FROM  grouped_data gd, total_sessions ts
ORDER BY gd.session_count DESC LIMIT 10;
```

---

**Original SQL**

---

```sql
WITH prep AS (
SELECT user_pseudo_id,
 (SELECT value.int_value FROM UNNEST(event_params) WHERE key = 'ga_session_id') AS session_id,
       ARRAY_AGG((SELECT value.string_value FROM UNNEST(event_params) WHERE key = 'source')
       IGNORE NULLS ORDER BY event_timestamp)[SAFE_OFFSET(0)] AS source,
       ARRAY_AGG((SELECT value.string_value FROM UNNEST(event_params) WHERE key = 'medium')
       IGNORE NULLS ORDER BY event_timestamp)[SAFE_OFFSET(0)] AS medium,
       ARRAY_AGG((SELECT value.string_value FROM UNNEST(event_params) WHERE key = 'campaign')
       IGNORE NULLS ORDER BY event_timestamp)[SAFE_OFFSET(0)] AS campaign
FROM `bigquery-public-data.ga4_obfuscated_sample_ecommerce.events_*`
WHERE _TABLE_SUFFIX BETWEEN '20201201' AND '20201231' GROUP BY user_pseudo_id, session_id )
grouped_data AS (
SELECT CASE
WHEN source = '(direct)' AND (medium IN ('(not set)', '(none)')) THEN 'Direct',
     WHEN REGEXP_CONTAINS(campaign, 'cross-network') THEN 'Cross-network'
WHEN (REGEXP_CONTAINS(source, 'alibaba|amazon|google shopping|shopify|etsy|ebay|stripe|walmart')
     OR REGEXP_CONTAINS(campaign, '^(.*(([^a-df-z]|^)shop|shopping).*)$'))
     AND REGEXP_CONTAINS(medium, '^(.*cp.*|ppc|paid.*)$') THEN 'Paid Shopping' ,
WHEN REGEXP_CONTAINS(source, 'baidu|bing|duckduckgo|ecosia|google|yahoo|yandex')
                     AND REGEXP_CONTAINS(medium, '^(.*cp.*|ppc|paid.*)$') THEN 'Paid Search' ,
WHEN REGEXP_CONTAINS(source, 'badoo|facebook|fb|instagram|linkedin|pinterest|tiktok|twitter|whatsapp')
                     AND REGEXP_CONTAINS(medium, '^(.*cp.*|ppc|paid.*)$') THEN 'Paid Social' ,
WHEN REGEXP_CONTAINS(source, 'dailymotion|disneyplus|netflix|youtube|vimeo|twitch|vimeo|youtube')
                     AND REGEXP_CONTAINS(medium, '^(.*cp.*|ppc|paid.*)$') THEN 'Paid Video' ,
WHEN medium IN ('display', 'banner', 'expandable', 'interstitial', 'cpm') THEN 'Display'
WHEN REGEXP_CONTAINS(source, 'alibaba|amazon|google shopping|shopify|etsy|ebay|stripe|walmart')
     OR REGEXP_CONTAINS(campaign, '^(.*(([^a-df-z]|^)shop|shopping).*)$') THEN 'Organic Shopping'
WHEN REGEXP_CONTAINS(source, 'badoo|facebook|fb|instagram|linkedin|pinterest|tiktok|twitter|whatsapp')
     OR medium IN ('social', 'social-network', 'social-media', 'sm', 'social network', 'social media') THEN 'Organic Social'
WHEN REGEXP_CONTAINS(source, 'dailymotion|disneyplus|netflix|youtube|vimeo|twitch|vimeo|youtube')
                     OR REGEXP_CONTAINS(medium, '^(.*video.*)$') THEN 'Organic Video'
WHEN REGEXP_CONTAINS(source, 'baidu|bing|duckduckgo|ecosia|google|yahoo|yandex') OR medium = 'organic'  THEN 'Organic Search'
WHEN REGEXP_CONTAINS(source, 'email|e-mail|e_mail|e mail') OR REGEXP_CONTAINS(medium, 'email|e-mail|e_mail|e mail') THEN 'Email'
WHEN medium = 'affiliate' THEN 'Affiliates' WHEN medium = 'referral' THEN 'Referral'
WHEN medium = 'audio' THEN 'Audio' WHEN medium = 'sms' THEN 'SMS'
WHEN medium LIKE '%push' OR REGEXP_CONTAINS(medium, 'mobile|notification') THEN 'Mobile Push Notifications' ELSE 'Unassigned'
END AS channel_grouping_session, COUNT(DISTINCT CONCAT(user_pseudo_id, session_id)) AS session_count
FROM
 prep GROUP BY channel_grouping_session
),
ORDER BY
 COUNT(DISTINCT CONCAT(user_pseudo_id, session_id)) DESC
LIMIT 1
```

Table 14: Citibike and taxi of NYC public data example, *condition parameters* surface rewrite.

| Question |
| --- |

For the top 20 Citi Bike routes in 2016, which route is faster than yellow taxis and among those, which one has the longest average bike duration? Please provide the start station name of this route. The coordinates are rounded to three decimals.

| Reference Plan |
| --- |

1. Focus on 2016 data to determine the top 20 most popular bike routes based on start and end stations, noting their latitude and longitude. 2. Calculate the average bike duration and count the number of bike trips for each route. 3. Extract the average duration for corresponding taxi routes using the same latitude and longitude for start and end points. 4. Calculate the average taxi duration for the matching routes. 5. Filter the results to find the bike route where the average bike duration is shorter than the average taxi duration. 6. Order the results by average bike duration in descending order and limit the output to one record.

| Gold SQL (After rewriting) |
| --- |

```sql
WITH top20route AS (
SELECT
 start_station_name, end_station_name, avg_bike_duration, avg_taxi_duration
FROM (
SELECT start_station_name, end_station_name,
 ROUND(start_station_latitude, 3) AS ss_lat, ROUND(start_station_longitude, 3) AS ss_long,
 ROUND(end_station_latitude, 3) AS es_lat, ROUND(end_station_longitude, 3) AS es_long,
 AVG(tripduration) AS avg_bike_duration,  COUNT(*) AS bike_trips
FROM
`bigquery-public-data.new_york.citibike_trips`
WHERE
 EXTRACT(YEAR from starttime) = 2015 AND start_station_name != end_station_name
GROUP BY  start_station_name, end_station_name, ss_lat, ss_long, es_lat, es_long
ORDER BY  bike_trips DESC LIMIT 20
) a
JOIN (
SELECT
 ROUND(pickup_latitude, 3) AS pu_lat,  ROUND(pickup_longitude, 3) AS pu_long,
 ROUND(dropoff_latitude, 3) AS do_lat,  ROUND(dropoff_longitude, 3) AS do_long,
 AVG(UNIX_SECONDS(dropoff_datetime)-UNIX_SECONDS(pickup_datetime)) AS avg_taxi_duration,
 COUNT(*) AS taxi_trips
FROM
`bigquery-public-data.new_york.tlc_yellow_trips_2015`
GROUP BY
 pu_lat, pu_long, do_lat, do_long
) b
ON
 a.ss_lat = b.pu_lat AND  a.es_lat = b.do_lat AND
 a.ss_long = b.pu_long AND  a.es_long = b.do_long
)
SELECT start_station_name FROM top20route
WHERE avg_bike_duration < avg_taxi_duration
ORDER BY avg_bike_duration DESC LIMIT 1
```

| Original SQL |
| --- |

```sql
SELECT
 start_station_name, end_station_name, avg_bike_duration, avg_taxi_duration, avg_taxi_fare
FROM (
SELECT start_station_name, end_station_name,
 ROUND(start_station_latitude, 3) AS ss_lat, ROUND(start_station_longitude, 3) AS ss_long,
 ROUND(end_station_latitude, 3) AS es_lat,ROUND(end_station_longitude, 3) AS es_long,
 COUNT(*) AS bike_trips
FROM`bigquery-public-data.new_york.citibike_trips`
WHERE start_station_name != end_station_name
GROUP BY start_station_name, end_station_name, ss_lat, ss_long, es_lat, es_long
ORDER BY bike_trips DESC LIMIT 100 ) a
JOIN (    SELECT
 ROUND(pickup_latitude, 3) AS pu_lat, ROUND(pickup_longitude, 3) AS pu_long,
 ROUND(dropoff_latitude, 3) AS do_lat, ROUND(dropoff_longitude, 3) AS do_long,
 COUNT(*) AS taxi_trips
FROM`bigquery-public-data.new_york.tlc_yellow_trips_2016`
GROUP BY pu_lat, pu_long, do_lat, do_long)b
ON
 a.ss_lat=b.pu_lat AND a.es_lat=b.do_lat AND a.ss_long=b.pu_long AND a.es_long=b.do_long
ORDER BY bike_trips DESC LIMIT 20;
```

Table 15: Google patent example, *advanced calculation* surface rewrite.

| Question |
| --- |

What is the publication number of US patent granted at January 2018, with the highest originality score based on the diversity of 4-digits IPC codes from its backward citations?

| Reference Plan |
| --- |

1. Filter US Patents: Select publication numbers and application numbers from the dataset, including only records where the country code is 'US', the grant date is within January 2018, and excluding records with a grant date of 0. Additionally, only consider patents with a specific kind code pattern (e.g., %B2%). 2. Extract IPC Codes: Select the publication number and count the unique 4-digit IPC codes associated with each selected patent. 3. Identify Maximum IPC Code Count: Create a subset of records that have the maximum count of a specific 4-digit IPC code for each patent. 4. Calculate IPC Occurrences in Backward Citations: Join the filtered patents with their backward citations. For each backward citation, join with the subset of records to get the 4-digit IPC codes, counting occurrences of each IPC code in the backward citations for each patent. 5. Compute Originality Score: For each patent, calculate an originality score based on the diversity of the 4-digit IPC codes from the backward citations, using a formula that considers the sum of squared occurrences of each IPC code, normalized by the total number of occurrences. 6. Select Highest Originality Score: From the computed originality scores, select the patent with the highest score. 7. Return Result: Output the publication number of the patent with the highest originality score.

| Gold SQL (After rewriting) |
| --- |

```sql
WITH patents_sample AS (
SELECT
 t1.publication_number, t1.application_number  FROM `patents-public-data.patents.publications` t1
WHERE
 country_code = 'US' AND grant_date between 20180101 AND 20180131
AND grant_date != 0 AND publication_number LIKE '%B2%'
),
interim_table AS (
SELECT t1.publication_number, SUBSTR(ipc_u.code, 0, 4) AS ipc4,
COUNT(SUBSTR(ipc_u.code, 0, 4)) AS ipc4_count
FROM
 patents-public-data.patents.publications t1, UNNEST(ipc) AS ipc_u
GROUP BY t1.publication_number, ipc4
),
chosen_ipc4_view AS (
        SELECT * FROM interim_table
        WHERE CONCAT(interim_table.publication_number, interim_table.ipc4_count) IN (
            SELECT CONCAT(publication_number, MAX(ipc4_count)) FROM interim_table GROUP BY  publication_number )
        ORDER BY ipc4_count DESC ),
ipc_counts AS (
        SELECT t1.publication_number, t3.ipc4, COUNT(t3.ipc4) AS ipc_occurrences
        FROM  patents_sample t1
        LEFT JOIN (
        SELECT
         x2.publication_number AS citing_publication_number, citation_u.publication_number AS backward_citation
        FROM
         patents-public-data.patents.publications x2, UNNEST(citation) AS citation_u) t2
        ON
         t2.citing_publication_number = t1.publication_number
        LEFT JOIN chosen_ipc4_view t3 ON  t3.publication_number = t2.backward_citation
        GROUP BY t1.publication_number, t3.ipc4
),
max_originality AS (
        SELECT publication_number,
        1 - SUM(POWER(ipc_occurrences, 2)) / POWER(SUM(ipc_occurrences), 2) AS originality
        FROM ipc_counts GROUP BY publication_number
        HAVING  SUM(ipc_occurrences) > 0 ORDER BY originality DESC LIMIT 1
)
SELECT  publication_number FROM max_originality
```

| Original SQL |
| --- |

```sql
WITH patents_sample AS (
SELECT
 t1.publication_number, t1.application_number  FROM `patents-public-data.patents.publications` t1
WHERE
 country_code = 'US' AND grant_date between 20180101 AND 20180131
AND grant_date != 0 AND publication_number LIKE '%B2%'
),
interim_table AS (
SELECT t1.publication_number, SUBSTR(ipc_u.code, 0, 4) AS ipc4,
COUNT(SUBSTR(ipc_u.code, 0, 4)) AS ipc4_count
FROM
 patents-public-data.patents.publications t1, UNNEST(ipc) AS ipc_u
GROUP BY t1.publication_number, ipc4
),
chosen_ipc4_view AS (
        SELECT * FROM interim_table
        WHERE CONCAT(interim_table.publication_number, interim_table.ipc4_count) IN (
            SELECT CONCAT(publication_number, MAX(ipc4_count)) FROM interim_table GROUP BY  publication_number )
        ORDER BY ipc4_count DESC ),
SELECT t1.publication_number, t3.ipc4, COUNT(t3.ipc4) AS ipc_occurrences
FROM  patents_sample t1
LEFT JOIN (
        SELECT
                 x2.publication_number AS citing_publication_number,
                citation_u.publication_number AS backward_citation
        FROM
         patents-public-data.patents.publications x2, UNNEST(citation) AS citation_u) t2
ON
 t2.citing_publication_number = t1.publication_number
LEFT JOIN chosen_ipc4_view t3 ON  t3.publication_number = t2.backward_citation
GROUP BY t1.publication_number, t3.ipc4
```

Table 16: Google analytics page conversion rate example, *advanced requirements* semantic rewrite.

**Question**

Calculate the conversion rate from product list pages to product detail pages for all sessions at January 2nd, 2021.

**Reference Plan**

1. query the event data to retrieve all unique event names 2. Selects events data from the Google Analytics 4 (GA4) sample e-commerce dataset for the specific date (20210102) 3. Filter to include only events named page_view, which represent page views. 4. flatten the nested event_params array and extract values for ga_session_id, ga_session_number, page_title, and page_location. This allows the analysis of individual page views within each user's session. 5. Further processes the unnested event data to classify pages based on URL depth and specific keywords into either Product Detail Pages (PDP) or Product Listing Pages (PLP). 6. Applies window functions to the categorized data to calculate the previous and next pages for each session per user, facilitating analysis of navigation paths between pages. 7. Filters sessions where the current page is a PLP and the next page is a PDP. 8. Counts the number of sessions transitioning from PLP to PDP and divides this by the total views of PLP pages to calculate the conversion rate.

**Gold SQL (After rewriting)**

```sql
WITH base_table AS (
SELECT
 event_name, event_date, event_timestamp, user_pseudo_id, user_id, device, geo, traffic_source, event_params, user_properties
FROM
`bigquery-public-data.ga4_obfuscated_sample_ecommerce.events_*`
WHERE
 _table_suffix BETWEEN'20210101'AND'20210131'AND event_name IN ('page_view')
),
unnested_events AS (
SELECT
        event_date ASdate, event_timestamp AS event_timestamp_microseconds,user_pseudo_id,
        MAX(CASE WHEN c.key = 'ga_session_id' THEN c.value.int_value END) AS visitID,
         MAX(CASE WHEN c.key = 'ga_session_number' THEN c.value.int_value END) AS visitNumber,
        MAX(CASE WHEN c.key = 'page_title' THEN c.value.string_value END) AS page_title,
        MAX(CASE WHEN c.key = 'page_location'THEN c.value.string_value END) AS page_location
FROM   base_table, UNNEST(event_params) c
GROUP BY 1, 2, 3 ),
unnested_events_categorised AS (
SELECT
 *,
CASE
WHEN ARRAY_LENGTH(SPLIT(page_location, '/')) >= 5 AND CONTAINS_SUBSTR(ARRAY_REVERSE(SPLIT(page_location, '/'))[SAFE_OFFSET(0)], '+')
AND (LOWER(SPLIT(page_location, '/')[SAFE_OFFSET(4)])
IN('accessories','apparel','brands','campus+collection','drinkware','electronics','google+redesign',
'lifestyle','nest','new+2015+logo','notebooks+journals','office','shop+by+brand','small+goods','stationery','wearables')
         OR LOWER(SPLIT(page_location, '/')[SAFE_OFFSET(3)]) IN
('accessories','apparel','brands','campus+collection','drinkware','electronics','google+redesign','lifestyle','nest','new+2015+logo',
notebooks+journals','office','shop+by+brand','small+goods','stationery','wearables')) THEN 'PDP'
WHEN NOT CONTAINS_SUBSTR(ARRAY_REVERSE(SPLIT(page_location, '/'))[SAFE_OFFSET(0)], '+') AND (LOWER(SPLIT(page_location,
'/')[SAFE_OFFSET(4)]) IN
('accessories','apparel','brands','campus+collection','drinkware','electronics','google+redesign','lifestyle','nest','new+2015+logo',
notebooks+journals','office','shop+by+brand','small+goods','stationery','wearables') OR LOWER(SPLIT(page_location,
'/')[SAFE_OFFSET(3)]) IN
('accessories','apparel','brands','campus+collection','drinkware','electronics','google+redesign','lifestyle','nest','new+2015+logo',
notebooks+journals','office','shop+by+brand','small+goods','stationery','wearables')) THEN 'PLP'
ELSE 'Other'                            END  AS page_category FROM unnested_events )
ranked_screens AS (
SELECT   *,
 LAG(page_category, 1) OVER (PARTITIONBY user_pseudo_id, visitID ORDER BY event_timestamp_microseconds) AS previous_page,
 LEAD(page_category, 1) OVER (PARTITIONBY user_pseudo_id, visitID ORDER BY event_timestamp_microseconds) AS next_page
FROM unnested_events_categorised
),
PLPtoPDPTransitions AS ( SELECT page_title, COUNT(*) AS transitions FROM ranked_screens WHERE page_category = 'PLP' AND next_page =
'PDP' GROUP BY page_title ),
TotalPLPViews AS ( SELECT page_title, COUNT(*) AS total_plp_views FROM ranked_screens WHERE page_category = 'PLP' GROUP BY
page_title ),
ConversionRates AS (
SELECT p.page_title, (CAST(t.transitions AS FLOAT64) / p.total_plp_views) * 100.0 AS conversion_rate
FROM TotalPLPViews p JOIN PLPtoPDPTransitions t ON p.page_title = t.page_title )
SELECT page_title, MAX(conversion_rate) AS max_conversion_rate FROM ConversionRates GROUP BY page_title ORDER BY max_conversion_rate
DESC LIMIT 1;
```

**Original SQL**

```sql
WITH base_table AS (
SELECT
 event_name, event_date, event_timestamp, user_pseudo_id, user_id, device, geo, traffic_source, event_params, user_properties
FROM
`bigquery-public-data.ga4_obfuscated_sample_ecommerce.events_*`
WHERE
 _table_suffix BETWEEN'20210101'AND'20210131'AND event_name IN ('page_view')
),
unnested_events AS (
SELECT
        event_date ASdate, event_timestamp AS event_timestamp_microseconds,user_pseudo_id,
        MAX(CASE WHEN c.key = 'ga_session_id' THEN c.value.int_value END) AS visitID,
         MAX(CASE WHEN c.key = 'ga_session_number' THEN c.value.int_value END) AS visitNumber,
        MAX(CASE WHEN c.key = 'page_title' THEN c.value.string_value END) AS page_title,
        MAX(CASE WHEN c.key = 'page_location'THEN c.value.string_value END) AS page_location
FROM   base_table, UNNEST(event_params) c
GROUP BY 1, 2, 3 ),
unnested_events_categorised AS (
SELECT
 *,
CASE
WHEN ARRAY_LENGTH(SPLIT(page_location, '/')) >= 5 AND CONTAINS_SUBSTR(ARRAY_REVERSE(SPLIT(page_location, '/'))[SAFE_OFFSET(0)], '+')
AND (LOWER(SPLIT(page_location, '/')[SAFE_OFFSET(4)])
IN('accessories','apparel','brands','campus+collection','drinkware','electronics','google+redesign',
'lifestyle','nest','new+2015+logo','notebooks+journals','office','shop+by+brand','small+goods','stationery','wearables')
         OR LOWER(SPLIT(page_location, '/')[SAFE_OFFSET(3)]) IN
('accessories','apparel','brands','campus+collection','drinkware','electronics','google+redesign','lifestyle','nest','new+2015+logo
','notebooks+journals','office','shop+by+brand','small+goods','stationery','wearables')) THEN 'PDP'
WHEN NOT CONTAINS_SUBSTR(ARRAY_REVERSE(SPLIT(page_location, '/'))[SAFE_OFFSET(0)], '+') AND (LOWER(SPLIT(page_location,
'/')[SAFE_OFFSET(4)]) IN
('accessories','apparel','brands','campus+collection','drinkware','electronics','google+redesign','lifestyle','nest','new+2015+logo
','notebooks+journals','office','shop+by+brand','small+goods','stationery','wearables') OR LOWER(SPLIT(page_location,
'/')[SAFE_OFFSET(3)]) IN
('accessories','apparel','brands','campus+collection','drinkware','electronics','google+redesign','lifestyle','nest','new+2015+logo
','notebooks+journals','office','shop+by+brand','small+goods','stationery','wearables')) THEN 'PLP'
ELSE 'Other'                            END  AS page_category FROM unnested_events )
SELECT (SELECT COUNT(*) FROM unnested_events_categorised WHERE page_title_adjusted='PDP')
/ (SELECT COUNT(*) FROM unnested_events_categorised)*100;
```

Table 17: GSOD and NYC public data example, *merge related SQLs* semantic rewrite.

---

### Question

Get the average number of trips on rainy and non-rainy days in New York City during 2016, using data from the closest weather station located near the coordinates (-74.0060, 40.7128). Define a "rainy day" as any day where the precipitation recorded is more than 0 millimeters.

### Reference Plan

1. Which days were rainy in 2016, and how can we obtain weather information? 2. The GHCN-D database allows us to access weather data from each weather station. 3. Given that the central coordinates of New York City are (-74.0060, 40.7128), we need to select a weather station to represent the weather data for New York City. 4. Calculate the weather stations closest to the center of New York City based on their distance. 5. Obtain the precipitation data from that weather station. 6. Use the precipitation data to classify the days in 2016 as either rainy or non-rainy. 7. The New York Citibike database stores daily bike rental data, which can be grouped based on whether it was a rainy day and then averaged. 8. Compare the differences in the average number of bike rentals on rainy days versus non-rainy days.

### Gold SQL (After rewriting)

```
WITH params AS (
SELECT ST_GeogPoint(-74.0060, 40.7128) AS center, 50 AS maxn_stations, 50 AS maxdist_km
),
distance_from_center AS (
  SELECT id, name, state,
    ST_GeogPoint(longitude, latitude) AS loc,
    ST_Distance(ST_GeogPoint(longitude, latitude), params.center) AS dist_meters
  FROM `bigquery-public-data.ghcn_d.ghcnd_stations`,
    params
 WHERE ST_DWithin(ST_GeogPoint(longitude, latitude), params.center, params.maxdist_km * 1000)
),
nearest_stations AS (
  SELECT *, RANK() OVER (ORDER BY dist_meters ASC) AS rank FROM  distance_from_cent
),
nearest_nstations AS (
SELECT
  station.* FROM nearest_stations AS station, params  WHERE rank <= params.maxn_stations ),
station_ids AS (
SELECT id, dist_meters from nearest_nstations ORDER BY dist_meters ASC LIMIT 50 ),
bicycle_rentals AS (
SELECT
    COUNT(starttime) as num_trips, EXTRACT(DATEfrom starttime) as trip_date
FROM`bigquery-public-data.new_york_citibike.citibike_trips` GROUP BY trip_date
),
closest AS (
SELECT station_ids.id as id, ANY_VALUE(station_ids.dist_meters) as dist
FROM `bigquery-public-data.ghcn_d.ghcnd_2016`AS wx
JOIN station_ids on wx.id=station_ids.id GROUP BY station_ids.id ORDER BY dist ASC LIMIT 1
),
rainy_days AS
(
SELECT date, COALESCE(MAX(prcp), 0) > 0 AS rainy
FROM (
SELECT wx.date AS date, IF (wx.element = 'PRCP', wx.value/10, NULL) AS prcp
FROM
`bigquery-public-data.ghcn_d.ghcnd_2016`AS wx
WHERE wx.id in (SELECT id FROM closest) ) GROUP BY date )
SELECT
 ROUND(AVG(bk.num_trips)) AS num_trips,  wx.rainy
FROM bicycle_rentals AS bk JOIN rainy_days AS wx ON wx.date = bk.trip_date GROUP BY wx.rainy
```

### Original SQL

```
--SQL1: New York City Rainy Days
WITH bicycle_rentals AS (
SELECT
        COUNT(starttime) as num_trips,
        EXTRACT(DATE from starttime) as trip_date
FROM`bigquery-public-data.new_york_citibike.citibike_trips` GROUP BY trip_date
),
rainy_days AS
( SELECT date, (MAX(prcp) > 5) AS rainy
      FROM (
              SELECT  wx.date ASdate,
              IF (wx.element = 'PRCP', wx.value/10, NULL) AS prcp
              FROM `bigquery-public-data.ghcn_d.ghcnd_2016`AS wx
              WHERE  wx.id = 'USW00094728'
        ) GROUP BY date   )
SELECT  ROUND(AVG(bk.num_trips)) AS num_trips, wx.rainy
FROM bicycle_rentals AS bk JOIN rainy_days AS wx
ON wx.date = bk.trip_date GROUP BY wx.rainy

--SQL2: Chicago Nearest Weather Station
WITH params AS (
SELECT ST_GeogPoint(-87.63, 41.88) AS center,
50 AS maxn_stations, 50 AS maxdist_km ),
distance_from_center AS (
SELECT
 id, name, state, ST_GeogPoint(longitude, latitude) AS loc,
 ST_Distance(ST_GeogPoint(longitude, latitude), params.center) AS dist_meters
FROM
`bigquery-public-data.ghcn_d.ghcnd_stations`,
 params
WHERE ST_DWithin(ST_GeogPoint(longitude, latitude),
params.center, params.maxdist_km*1000)
)
SELECT * from distance_from_center
```

---

### B.3 SPIDER 2.0 DATABASE EXAMPLES

Google Analytics 4 serves as a notable example of a Spider 2.0 database (see Fig. 11). For each Google Analytics 4 property and linked Firebase project enabled for BigQuery export, a dataset named analytics_{property_id} is created. Within the dataset, daily tables named events_YYYYMMDD are generated when the Daily export option is enabled.

To accommodate latency, Google Analytics 4 updates these daily tables for up to three days with late-arriving events, ensuring proper timestamping. Each column in these tables represents specific event parameters, some of which are nested within RECORDS and may be repeated. For instance, the item_params RECORD stores custom item parameters unique to each implementation.

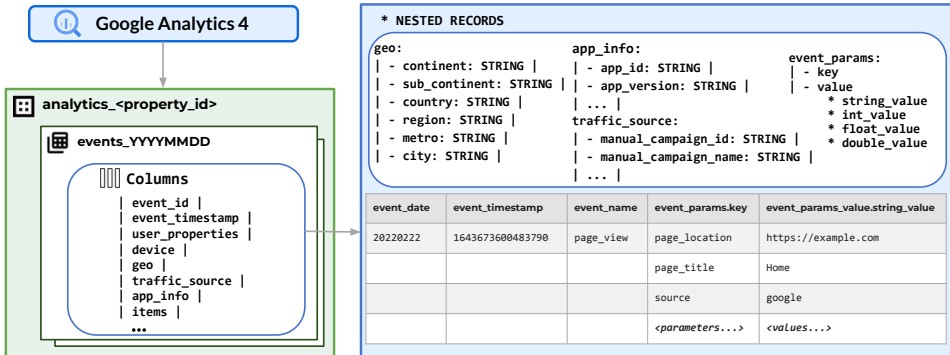

Figure 11: Google analytics 4 database schema with nested record.

Fig. 12 showcases an example of an enterprise-level real-world database environment from Spider 2.0, with multiple schemas to navigate through, each of them containing a variety of tables. It highlights the complex structure types of Spider 2.0 databses, which exemplifies how our benchmark encompasses a broader and more intricate variety compared to others.

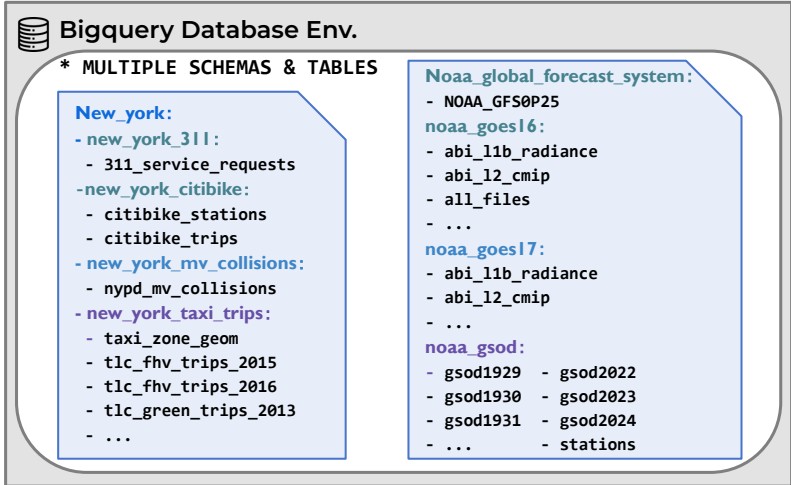

Figure 12: Bigquery database environment with multiple schema and tables.

### B.4 EXAMPLES OF EXTERNAL DOCUMENTS

In this section, we present the external documents utilized in Spider 2.0. The first is a table that outlines the categorization method for traffic channels. The original document provided an HTML table, which we present here in Fig. 13. The second document is the Google Page Category as shown in Fig. 14, which demonstrates how to classify a page into categories such as Product List Page and Product Detail Page.

```
| Channel                  | Conditions

|-------------------------|---------------------------------------------------------------------------------------------------------------
--------------------|
| **Direct**              | Source exactly matches "(direct)"
AND
Medium is one of ("(not set)", "(none)")

| **Cross-network**       | Campaign Name contains "cross-network"
Cross-network includes Demand Gen, Performance Max and Smart Shopping.

| **Paid Shopping**       | Source matches a list of shopping sites (alibaba, amazon, google shopping, shopify, etsy, ebay, stripe, walmart)
OR
Campaign Name
df-z]\|^)shop\|shopping).*)$`
AND
Medium matches regex `^(.*cp.*\|ppc\|retargeting\|paid.*)$` |
| **Paid Search**         | Source matches a list of search sites (baidu,bing,duckduckgo,ecosia,google,yahoo,yandex)
AND
Medium matches regex `^(.*cp.*\|ppc\|
| **Paid Social**         | Source matches a regex list of social sites (badoo,facebook,fb,instagram,linkedin,pinterest,tiktok,twitter,whatsapp)
AND
Medium ma
`^(.*cp.*\|ppc\|retargeting\|paid.*)$`                                                                                  |
| **Paid Video**          | Source matches a list of video sites (dailymotion,disneyplus,netflix,youtube,vimeo,twitch,vimeo,youtube)
AND
Medium matches regex
`^(.*cp.*\|ppc\|retargeting\|paid.*)$`                                                                   |
| **Display**             | Medium is one of ("display", "banner", "expandable", "interstitial", "cpm")

| **Organic Shopping**    | Source matches a list of shopping sites (alibaba,amazon,google shopping,shopify,etsy,ebay,stripe,walmart)
OR
Campaign name matches
z]\|^)shop\|shopping).*)$`                                                                                               |
| **Organic Social**      | Source matches a regex list of social sites (badoo,facebook,fb,instagram,linkedin,pinterest,tiktok,twitter,whatsapp)
OR
Medium is
"social-network", "social-media", "sm", "social network", "social media")                                                |
| **Organic Video**       | Source matches a list of video sites (dailymotion,disneyplus,netflix,youtube,vimeo,twitch,vimeo,youtube)
OR
Medium matches regex `
| **Organic Search**      | Source matches a list of search sites (baidu,bing,duckduckgo,ecosia,google,yahoo,yandex)
OR
Medium exactly matches organic

| **Referral**            | Medium exactly matches Referral                                                                                        |
| **Email**               | Source = email\|e-mail\|e_mail\|e mail
OR
Medium = email\|e-mail\|e_mail\|e mail

| **Affiliates**          | Medium = affiliate

| **Audio**               | Medium exactly matches audio

| **SMS**                 | Source exactly matches sms
OR
Medium exactly matches sms

| **Mobile Push Notifications** | Medium ends with "push"
OR
Medium contains "mobile" or "notification"
| **Unassigned** | Others                              |
```

Figure 13: Channel group of Google Analytics, external document for a bigquery example.

---

**Product page category**

### Refined Page Classification Criteria

#### Overview
To enhance our understanding of user engagement on our e-commerce platform, we differentiate between two types of pages based on the URL structure: Product Listing Pages (PLPs) and Product Detail Pages (PDPs). These classifications are crucial for analyzing user behavior and improving site navigation efficiency.

#### Product Listing Pages (PLPs)
PLPs are identified by specific characteristics in the URL:
- The URL must be divided into at least five segments.
- Neither the fourth nor the fifth segment contains a '+' sign, ensuring these are not detail views.
- The fourth or fifth segment must contain one of the following category names, indicating a broader category or collection page rather than a specific product focus:
- Accessories - Apparel - Brands - Campus Collection - Drinkware - Electronics - Google Redesign - Lifestyle - Nest - New 2015 Logo - Notebooks & Journals - Office - Shop by Brand - Small Goods - Stationery - Wearables

#### Product Detail Pages (PDPs)
PDPs, which focus on individual products, are marked by:
- A URL split into at least five segments, akin to PLPs.
- The presence of a '+' sign in the last segment, a common marker for detailed product pages.
- The fourth or fifth segment must also include one of the specified category names, ensuring that the detail being viewed pertains to one of the recognized product categories:
- Accessories - Apparel - Brands - Campus Collection - Drinkware - Electronics - Google Redesign - Lifestyle - Nest - New 2015 Logo - Notebooks & Journals - Office - Shop by Brand - Small Goods - Stationery - Wearables

### Conclusion
This detailed classification approach enables a more nuanced analysis of user pathways and interactions on our platform. By distinguishing between general browsing (PLPs) and targeted product interest (PDPs), we can tailor our content and design strategies to better meet the needs of our users, ultimately enhancing the shopping experience and improving business outcomes.

Figure 14: Page category document of Google analytics 4.

## B.5 EXAMPLES OF CONTEXT SETUP

Besides the context setup method for the DBT project mentioned in App.B.2, we will outline the process for establishing the context in a example about query database.

For the task, *Can you provide the details of the top 5 longest bike share trips that started during the second half of 2017?*, 'query.py' serves as our predefined interface for interaction between the model and the cloud database. This question is inherently ambiguous; without specifying the answer format constraints, evaluating the responses becomes challenging. Therefore, we provide "result.csv", which defines the required answer format.

```
|--- README.md   # The task description
|--- query.py    # The query interface
|--- bigquery_credential.json   # Bigquery credentials
'--- result.csv  # Answer format of data in November 2022

-- result.csv
trip_id,duration_sec,star_date,start_station_name,route,bike_number,
subscriber_type,member_birth_year,age,age_class,member_gender,region_name
```

For the examples presented in Tab. 13, we outline the setup details for the Spider 2.0 example. Additionally, we provide answer examples for specific cases, which not only constrain the answer format but also enable the agent to perform self-debugging using these examples.

The task instruction is *Provide the number of sessions and percentage breakdown by channel for December 2020*. We supply '202011.csv' and '202101.csv' as demonstration answers. We envision a real SQL writing scenario where the agent can first query November 2020 to check for consistency with '202011.csv'. If discrepancies arise, the agent can identify that their SQL is incorrect and make the necessary corrections. Note that this is not a task requirement; it is simply our belief that real SQL writing has such a need, and we will not mandate that the model does this. We believe this approach reflects a natural and realistic setting, although we only provide answer constraints for a limited number of examples.

```
|--- README.md   # The task description
|--- query.py    # The query interface
|--- BASIC_SQLs  # SQL examples of google analytics
|--- bigquery_credential.json   # Bigquery credentials
|--- 202012.csv  # The predefined answer file,
|--- 202101.csv  # Answer format of data in January 2021
'--- 202011.csv  # Answer format of data in November 2022

-- 202011.csv

item_name,item_quantity
Google Decal,103
Google Clear Pen 4-Pack,81
Google Mesh Bag Red,79
Google Mini Kick Ball,77
Google Light Pen Red,8
Google Laptop and Cell Phone Stickers,7
Google Pen Neon Coral,7
Google Metallic Notebook Set,7
Google Pen Lilac,5
Google Pen Red,5
```

The query interface of Bigquery "query.py" is

```
import os
import pandas as pd
from google.cloud import bigquery

def query_data(sql_query, is_save, save_path="result.csv"):
    """
    Queries data from BigQuery based on the provided SQL query and handles the result.

    Args:
    sql_query (str): SQL query string to be executed.
    is_save (bool): If True, saves the query results to a CSV file at the specified save_path.
                    If False, prints the results to the console.
```

```
        save_path (str): The file path where the results will be saved if is_save is True.
        Defaults to 'result.csv'.
        """
        os.environ["GOOGLE_APPLICATION_CREDENTIALS"] = "bigquery_credential.json"
        client = bigquery.Client()
        query_job = client.query(sql_query)
        try:
          results = query_job.result().to_dataframe()
          if results.empty:
            print("No data found for the specified query.")
          else:
            if is_save:
                results.to_csv(save_path, index=False)
                print(f"Results saved to {save_path}")
            else:
                print(results)
        except Exception as e:
          print("Error occurred while fetching data: ", e)

if __name__ == "__main__":

    # Write your SQL query in the sql_query variable to interact with the database,
    #example SQL query related to this task is provided below
    sql_query = """
    SELECT
      *
    FROM
      `bigquery-public-data.ga4_obfuscated_sample_ecommerce.events_*`
    WHERE
      _TABLE_SUFFIX BETWEEN '20201201' AND '20201231'
    LIMIT 1
    """
    query_data(sql_query, is_save=True, save_path="result.csv")
```

## B.6 THE DIFFERENCE IN TASK INSTRUCTIONS BETWEEN SPIDER 2.0 AND SPIDER 2.0-LITE.

During the annotation process, we found that *unambiguity* and *naturalness* are two mutually exclusive concepts. Therefore, in Spider 2.0-Lite, we emphasize unambiguity, while in Spider 2.0, we emphasize naturalness. The two instructional approaches restore the possible question forms that may arise in real-world text-to-SQL workflows.

**Example 1:**

Spider 2.0: The company management has requested a detailed report on the year-to-date performance of the Magnificent 7 stocks.

Spider 2.0-lite: Please show the price change rate of the Magnificent 7 stocks from the beginning of this year to today.

**Example 2:**

Spider 2.0: Can you provide the details of the top 5 longest bike share trips that started during the second half of 2017?

Spider 2.0-lite: Can you provide the details of the top 5 longest bike share trips that started during the second half of 2017, including the trip ID, duration in seconds, start date, start station name, route (start station to end station), bike number, subscriber type, member's birth year, age, age classification, gender, and the region name of the start station?

**Example 3:**

Spider 2.0: What's the no-tip percentage for NYC yellow taxi trips in each borough from January 1-7, 2016, considering valid trips with at least one passenger and non-negative amounts?

Spider 2.0-lite: For NYC yellow taxi trips between January 1-7, 2016, could you tell me the percentage of no tips in each borough. Ensure trips where the dropoff occurs after the pickup, the passenger count is greater than 0, and trip distance, tip, tolls, MTA tax, fare, and total amount are non-negative.

## B.7 SQL DIALECT DOCUMENTS COLLECTION

The core of SQL dialects lies in different advanced functions and subtle syntax variations across SQL versions. To support retrieval-augmented agent frameworks, we crawled and pre-processed the function documents for different database systems from their official websites. The detailed statistics of the crawled web pages and parsed categories/functions are presented in Tab. 18. Note that, functions belonging to the same category (e.g., `aggregate` functions like `COUNTIF` and `STRING_AGG`) may be introduced in the same paragraph in some web pages. In this case, we re-use the description on this shared function category for different concrete functions.

Table 18: Statistics of different database systems on Spider 2.0. Notice that, [†] means there is no well-defined function list in the official web pages for Postgres, thus we merely use the summarized document for each function category.

| Database | Documentation Website | # Page | # Category | # Function |
|---|---|---|---|---|
| BigQuery | `https://cloud.google.com/bigquery/docs/reference/standard-sql/functions-and-operators` | 34 | 34 | 390 |
| Snowflake | `https://docs.snowflake.com/en/sql-reference/` | 719 | 30 | 719 |
| Postgres | `https://www.postgresql.org/docs/current/functions.html` | 30 | 30 | 30[†] |
| Clickhouse | `https://clickhouse.com/docs/en/sql-reference/functions` | 226 | 6 | 226 |
| SQLite | `https://www.sqlite.org/docs.html` | 6 | 6 | 147 |
| DuckDB | `https://duckdb.org/docs/sql/functions/overview` | 24 | 24 | 513 |
| **Total** | | 1039 | 130 | 2025 |

### B.7.1 PROCESSED FUNCTIONS FOR DIFFERENT DATABASE SYSTEMS

In this section, we demonstrate examples of parsed documents for different database systems. These pre-parsed chunks can be retrieved and inserted into the prompt to compensate agents for their deficiencies in SQL dialect knowledge.

**Document of BigQuery Functions**

> database="BigQuery", function="ST_INTERSECTS", category="geography-functions"
>
> ## ST_INTERSECTS
>
> ST_INTERSECTS(geography_1, geography_2)
>
> **Description**
>
> Returns ` TRUE ` if the point set intersection of ` geography_1 ` and ` geography_2 ` is non-empty. Thus, this function returns ` TRUE ` if there is at least one point that appears in both input ` GEOGRAPHY ` s.
>
> If ` ST_INTERSECTS ` returns ` TRUE ` , it implies that ` ST_DISJOINT ` returns ` FALSE ` .
>
> **Return type**
>
> ` BOOL `

**Document of Postgres Functions**

**database="Postgres", function_category="enum-support-functions"**

For enum types (described in Section 8.7), there are several functions that allow cleaner programming without hard-coding particular values of an enum type. These are listed in Table 9.35. The examples assume an enum type created as:

```
CREATE TYPE rainbow AS ENUM ("red", "orange", "yellow", "green",
"blue", "purple");
```

Table 9.35. Enum Support Functions

Function
Description
Example(s)

enum_first ( anyenum ) → anyenum
Returns the first value of the input enum type.
enum_first(null::rainbow) → red

enum_last ( anyenum ) → anyenum
Returns the last value of the input enum type.
enum_last(null::rainbow) → purple

enum_range ( anyenum ) → anyarray
Returns all values of the input enum type in an ordered array.
enum_range(null::rainbow) → red,orange,yellow,green,blue,purple

enum_range ( anyenum, anyenum ) → anyarray
Returns the range between the two given enum values, as an ordered array. The values must be from the same enum type. If the first parameter is null, the result will start with the first value of the enum type. If the second parameter is null, the result will end with the last value of the enum type.
enum_range("orange"::rainbow, "green"::rainbow) → orange,yellow,green
enum_range(NULL, "green"::rainbow) → red,orange,yellow,green
enum_range("orange"::rainbow, NULL) → orange,yellow,green,blue,purple

Notice that except for the two-argument form of enum_range, these functions disregard the specific value passed to them; they care only about its declared data type. Either null or a specific value of the type can be passed, with the same result. It is more common to apply these functions to a table column or function argument than to a hardwired type name as used in the examples.

**Document of Snowflake Functions**

---

database="Snowflake", function="`ATAN2`", category="numeric-functions"

Categories: Numeric functions (Trigonometric)

## ATAN2

Computes the inverse tangent (arc tangent) of the ratio of its two arguments.
For example, if $x > 0$, then the expression ATAN2(y, x) is equivalent to ATAN(y/x).
The arc tangent is the angle between:

The X axis.
The ray from the point (0,0) to the point (X, Y) (where X and Y are not both 0).

See also: ATAN

## Syntax

ATAN2( $<y>$ , $<x>$ )

Copy Note that the first parameter is the Y coordinate, not the X coordinate.

## Arguments

y This parameter is the Y coordinate of the point at the end of the ray. The data type is DOUBLE.

x This parameter is the X coordinate of the point at the end of the ray. The data type is DOUBLE.

## Returns

The data type of the returned value is DOUBLE.
The returned value is in radians, not degrees.
The returned value is a number in the interval [-pi, pi].

## Usage notes

If the data type of an argument is a numeric data type other than DOUBLE, then the value is converted to DOUBLE.
If the data type of an argument is string, the value is converted to DOUBLE if possible.
If the data type of an argument is any other data type, the function returns an error.
If either argument is NULL, the returned value is NULL.

## Examples

SELECT ATAN2(5, 5);

```
--------------+
ATAN2(5, 5)   |
--------------+
0.7853981634  |
--------------+
```

**Document of DuckDB Functions**

---

database="DuckDB", function="`datediff`", category="date-functions"

Function: datediff(part, startdate, enddate)
The number of partition boundaries between the dates. Alias of `date_diff`.

Description: The number of partition boundaries between the dates.
Example: `datediff('month', DATE '1992-09-15', DATE '1992-11-14')`
Result: 2
Alias: date_diff.

**Document of SQLite Functions**

---

**database="SQLite", function="`group_concat(X,Y)`", category="aggregate-functions"**

Function: group_concat(X,Y)

Usage: group_concat(X) group_concat(X,Y) string_agg(X,Y)

Descritpion: The group_concat() function returns a string which is the concatenation of all non-NULL values of X. If parameter Y is present then it is used as the separator between instances of X. A comma (`","`) is used as the separator if Y is omitted.

The string_agg(X,Y) function is an alias for group_concat(X,Y). String_agg() is compatible with Post-greSQL and SQL-Server and group_concat() is compatible with MySQL.

The order of the concatenated elements is arbitrary unless an ORDER BY argument is included immediately after the last parameter.

---

**Document of Clickhouse Functions**

---

**database="Clickhouse", function="`JSONHas`", category="json-functions"**

## JSONHas
If the value exists in the JSON document, 1 will be returned. If the value does not exist, 0 will be returned.

### Syntax

JSONHas(json [, indices_or_keys]...)

### Parameters
json
- JSON string to parse. String
indices_or_keys
- A list of zero or more arguments, each of which can be either string or integer. String, Int*.
indices_or_keys
type:
String = access object member by key.
Positive integer = access the n-th member/key from the beginning.
Negative integer = access the n-th member/key from the end.

### Returned value
Returns 1 if the value exists in json , otherwise 0. UInt8.

### Examples
Query:

SELECT JSONHas('{"a": "hello", "b": [-100, 200.0, 300]}', 'b') = 1
SELECT JSONHas('{"a": "hello", "b": [-100, 200.0, 300]}', 'b', 4) = 0

The minimum index of the element is 1. Thus the element 0 does not exist. You may use integers to access both JSON arrays and JSON objects. For example:

SELECT JSONExtractKey('{"a": "hello", "b": [-100, 200.0, 300]}', 1) = 'a'
SELECT JSONExtractKey('{"a": "hello", "b": [-100, 200.0, 300]}', 2) = 'b'
SELECT JSONExtractKey('{"a": "hello", "b": [-100, 200.0, 300]}', -1) = 'b'
SELECT JSONExtractKey('{"a": "hello", "b": [-100, 200.0, 300]}', -2) = 'a'
SELECT JSONExtractString('{"a": "hello", "b": [-100, 200.0, 300]}', 1) = 'hello'

---

## B.8    EXTEND DATASET STATISTIC

**Database scope.** As shown in Fig. 15, the databases utilized in Spider 2.0 encompass a wide array of domains and real-world scenarios, providing a notable degree of diversity.

**Data types.** As depicted in Fig. 16, the Spider 2.0 database encompasses a wide variety of data types, including text-based data types including STRING and BOOLEAN, number-based like INTEGER and FLOAT, structured data as STRUCT, JSON, time-related data such as TIMESTAMP, and spatial

data like `GEOGRAPHY` in google bigquery datasets. The diversity and breadth of these data types underscore the extensive complexity and wide-ranging nature of our benchmark database. This variability is reflected in the SQL dialects and the intricacies of data handling, thereby presenting significant challenges for SQL generation.

**Keywords.** As shown in Fig. 17, due to the complexity of the SQL in the Spider 2.0 dataset and its coverage of various dialects, it contains more SQL keywords than any previous datasets.

**Number of Tables.** As shown in Fig. 17, the databases in Spider 2.0 contain more tables than previous datasets. Additionally, each SQL query in Spider 2.0 requires joining more tables on average.

**Data Volume.** The databases used in Spider 2.0 contain significantly larger data volumes. In comparison, each database in WikiSQL has only 17 rows, Spider 1.0 contains 2K rows, KaggleDBQA has 280K rows, and BIRD has 549K rows. In contrast, the average database in Spider 2.0 has **5273.42M rows**, with many databases reaching **TB-level** sizes.

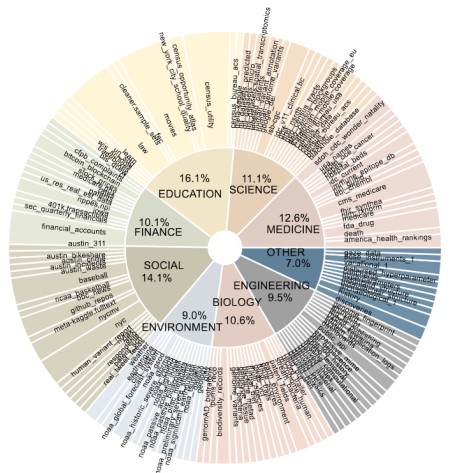

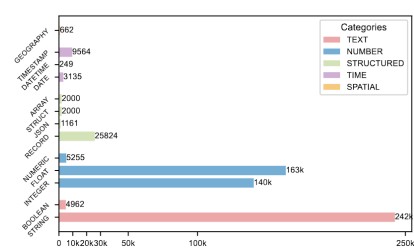

Figure 16: Data types of Spider 2.0 database.

Figure 15: Domain distribution of Spider 2.0 database.

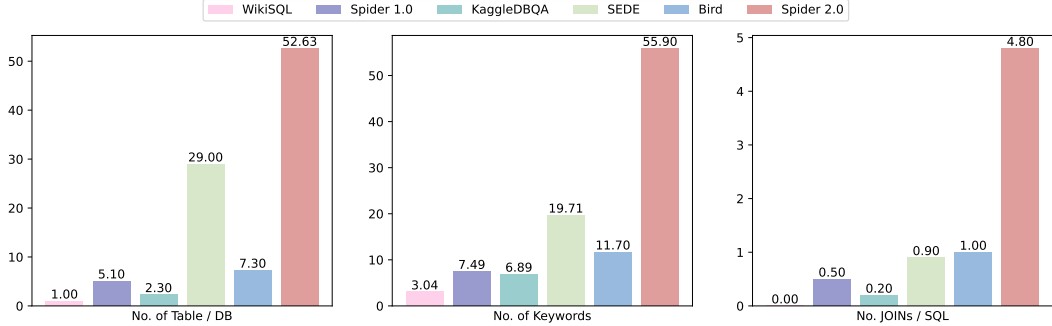

Figure 17: A comparative statistical analysis of SQL queries in Spider 2.0 and previous text-to-SQL benchmarks.

# C  DETAILS OF EXPERIMENTS

## C.1  SPIDER-AGENT FRAMEWORK

Inspired by React (Yao et al., 2022) and Intercode (Yang et al., 2023), we developed an agent framework called Spider-Agent, which is primarily focused on database-related coding tasks and projects.

**Spider 1.0**

```sql
SELECT T2.name, T2.budget
FROM instructor as T1 JOIN department as
T2 ON T1.department_id = T2.id
GROUP BY T1.department_id
HAVING avg(T1.salary) >
    (SELECT avg(salary) FROM instructor)
```

**KaggleDBQA**

```sql
SELECT T1.school_district
FROM FINREV_FED_17 as T1 JOIN
FINREV_FED_KEY_17 as T2
ON T1.state_code = T2.state_code WHERE
T2.state = "Wisconsin"
 ORDER BY T1.t_fed_rev DESC LIMIT 1
```

**SEDE**

```sql
SELECT  top 50 count(v.postid) as 'Total Votes',
v.postid AS [Post Link]
FROM Votes v
INNER JOIN Posts p ON p.id = v.postid
WHERE
PostTypeId = 1
AND v.VoteTypeId=2
AND p.tags like LOWER('%<' + ##tagname:string## + '>%')
AND p.CreationDate >= DATEADD(month, -6, GETDATE())
GROUP BY v.postid
ORDER BY 'Total Votes' desc
```

**BIRD**

```sql
SELECT coachID FROM coaches
WHERE lgID='NBA' AND post_wins !=0
AND post_losses != 0 AND coachID IN
(SELECT coachID FROM coaches WHERE lgID='NBA'
GROUP BY coachID
HAVING COUNT(tmID)>=2)
 ORDER BY post_losses ASC LIMIT 1;
```

**Spider 2.0**

```sql
WITH cohorts AS (
  SELECT
    User_id, DATE(created_at) AS order_date,
    DATE(DATE_TRUNC(FIRST_VALUE(created_at)
    OVER(PARTITION BY user_id ORDER BY created_at), month)) AS cohort
  FROM `bigquery-public-data.thelook_ecommerce.orders`
  WHERE DATE(created_at) BETWEEN '2020-01-01' AND '2020-12-31'
),
-- calculate the number of months after the first month
activity AS (
  SELECT
    User_id, cohort,
    DATE_DIFF(order_date, cohort, month) AS month_since_first_order
  FROM cohorts
  WHERE DATE(cohort) = '2020-01-01'
),
-- counting the number of unique users for each cohort new_users AS (
  SELECT
    cohort, month_since_first_order, COUNT(DISTINCT user_id) AS new_user
  FROM
    activity
  GROUP BY cohort, month_since_first_order
),
-- calculate the total customer on each cohort
cohort_users AS (
  SELECT
    cohort, month_since_first_order, new_user,
    FIRST_VALUE(new_user)
OVER(PARTITION BY cohort ORDER BY month_since_first_order) AS cohort_user
  FROM  new_users
)
-- calculate the cohort users percentage
SELECT
  cohort, month_since_first_order,
  new_user, cohort_user,
  new_user / cohort_user AS cohort_users_percentage
FROM
  cohort_users
WHERE
  month_since_first_order > 0 -- Exclude January data (month 0)
ORDER BY
  cohort,
    month_since_first_order;
```

Figure 18: A comparison of SQL examples selected based on the median token length for Spider 2.0 and previous text-to-SQL benchmarks. Spider 2.0 examples were selected with token counts at the median, while examples from the other four datasets were selected from the original papers.

The framework allows for multi-turn interactions with the database via command-line interfaces until the final answer is obtained. To ensure the agent focuses solely on interacting with the database, as shown in Tab. 19, we designed a set of specialized actions. We use a temperature of 1.0 and top-p of 0.9 and truncate from the beginning of the input if still exceeding the max tokens limit required by the models.

The model automatically terminates if it outputs the same result three times in a row or if any action takes longer than 120 seconds. The prompts used in the experiments are provided in App.C.7. We heuristically request the agents to complete the tasks within a max step limit of 30, which is enough for most tasks.

Table 19: The action space of Spider-Agent, an agent baseline method for Spider 2.0.

| Action | Description |
| --- | --- |
| BASH | Executes shell commands, such as checking file information, running code, or executing DBT commands. |
| CreateFile | Creates a new file with specified content. |
| EditFile | Edits or overwrites the content of an existing file. |
| ExecuteSQL | Executes a SQL query on BigQuery/Snowflake, with an option to print or save the results. |
| GetTables | Retrieves all table names and schemas from a specified BigQuery/Snowflake dataset. |
| GetTabInfo | Retrieves detailed column information for a specific table in BigQuery/Snowflake. |
| SampleRows | Samples a specified number of rows from a BigQuery/Snowflake table and saves them as JSON. |
| FAIL | Agent decides the task is infeasible. |
| Terminate | Agent decides the task is finished. |

**The number of joins does not have a direct correlation with model performance.** For Tab. 19, there was no clear correlation observed between performance and the number of joins. We speculate that this is due to the fact that during the SQL annotation process, we ensured that all examples were quite complex, which made performance independent of the number of tables in SQL involved.

**Action analysis of Spider-Agent.** We analyze the results of Spider-Agent. For all correctly completed tasks, the agent needed an average of 9.0 steps (with a maximum of 17 steps and a minimum of 6 steps) within the defined action space to achieve the desired result. We also analyze the frequency with which actions are invoked at each turn by Spider-Agent, as shown in Fig. 20.

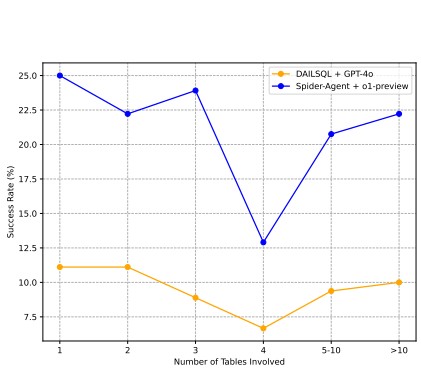

Figure 19: The effect of the number of *Join* on performance.

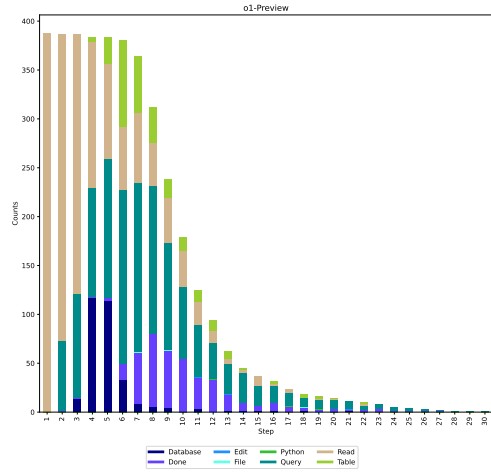

Figure 20: The frequency with which actions are invoked at each turn by Spider-Agent w/ o1-Preview for task instances that it solved on the Spider 2.0 (286 trajectories).

## C.2 DETAILS OF SPIDER 2.0-LITE EXPERIMENTS

**Details of baseline methods.** LLM-based text-to-SQL methods have demonstrated exceptional zero-shot reasoning and domain generalization capabilities. DIN-SQL (Pourreza & Rafiei, 2024)

employs task decomposition and adaptive prompting strategies tailored to task complexity. DAIL-SQL (Gao et al., 2024) achieves the best EX on Spider through elaborately designed prompt optimizations and in-context learning. CHESS (Talaei et al., 2024) integrates schema filtering based on entity and context retrieval, and SQL revision, achieving the best EX on BIRD. CodeS (Li et al., 2024a) fine-tunes open-source code generation models on extensive text-to-SQL corpora, obtaining performance comparable to methods that are based on prompting LLMs.

**The treatment for sampled cell values.** Spider 2.0-lite contains various complex data types, such as nested structures (`RECORTED`) or array (`REPRATED`) in BigQuery. if we only provide data type indicators, it is challenging for models to correctly process these types by appropriate SQL functions. Therefore, we provide sampled cell values (in markdown format) from each table in the prompt for all evaluated methods.

**The treatment for value linking.** During evaluation, we do not perform value linking (entity retrieval in CHESS, value retriever in CodeS) when solving instances from BigQuery, as the API cost of retrieving all values from a terabyte-scale cloud database is prohibitively expensive. Since value linking is crucial for identifying entities in filter conditions, its omission may hinder performance. Exploring cost-efficient methods for value linking or alternative approaches is an important direction for future work.

**LLM.** Given the extensive length of prompts after serializing large-scale schemas, we default to using GPT-4o, which supports a 128K context window, as the base LLM. Specifically for CHESS, we use GPT-3.5-turbo for column filtering to reduce costs.

**Temperature.** For all methods, we set the temperature of the LLM to 0 to ensure the reproducibility of the results.

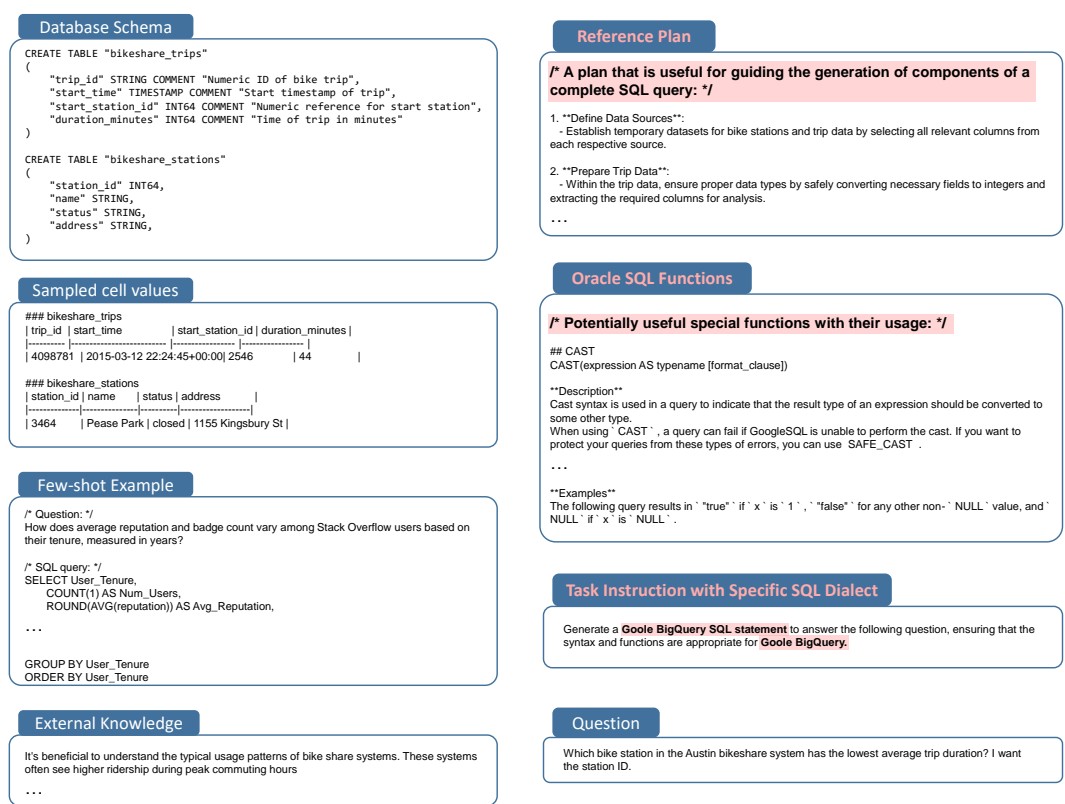

Figure 21: An example of prompt organization given by DAIL-SQL. prompt components that tailored to Spider 2.0-lite are highlighted. All these prompt components are similarly implemented for all other evaluated baseline methods including DIN-SQL, CHESS and CodeS.

## C.3 DETAILS OF ERROR ANALYSIS

We summarize the descriptions for all error categories in Fig. 22.

| Error Type | Subcategory | Description | Example of Predicted SQL | Example of Gold SQL |
|---|---|---|---|---|
| Wrong Schema Linking | Wrong Table | Some of the requested tables are incorrect or do not exist,or excessive tables are requested,or some ground truth tables are missing. | FROM spider2-public-data.wide_world_importers.sales_Customers c JOIN spider2-public-data.wide_world_importers.sales_Orders o ON c.CustomerID = o.CustomerID JOIN spider2-public-data.wide_world_importers.sales_Invoices i ON c.CustomerID = i.CustomerID | FROM `spider2-public-data.wide_world_importers.sales_Customers` cu INNER JOIN `spider2-public-data.wide_world_importers.sales_Orders` o ON cu.CustomerID = o.CustomerID INNER JOIN `spider2-public-data.wide_world_importers.sales_OrderLines` ol ON o.OrderID = ol.OrderID INNER JOIN `spider2-public-data.wide_world_importers.sales_Invoices` Inv ON ol.OrderID = Inv.OrderID ) Orders |
| | Wrong Column | Some of the requested columns are incorrect or do not exist,or excessive columns are requested,or some ground truth columns are missing. | WITH Yearly_Delivered_Orders AS ( SELECT strftime('%Y', order_purchase_timestamp) AS year, ... | WITH monthly_order_counts AS ( SELECT strftime('%Y', order_delivered_customer_date) AS Year ... |
| Erroneous Data Analysis | Incorrect Dialect Function Usage | Incorrect or missing use of dialect-specific functions for string manipulation (e.g., CONCAT), date processing (e.g., DATE_TRUNC), or geographic data (e.g., ST_DISTANCE), etc. | SELECT ... (LAG(longitude, 1) OVER (ORDER BY iso_time ASC) - longitude) AS lon_diff, (LAG(latitude, 1) OVER (ORDER BY iso_time ASC) - latitude) AS lat_diff FROM hurricane_data | SELECT ... ST_DISTANCE(geom, LAG(geom, 1) OVER ( BY sid ORDER BY iso_time ASC)) / 1000 AS dist FROM hurricane_geometry |
| | Incorrect Data Calculation | Advanced data calculations fail to meet the intended outcomes, often due to errors in grouping, aggregation (e.g., AVG, SUM), window functions (e.g., PARTITION BY, NTILE), or formula application (e.g., CORR, STDDEV). | Refer to Fig. 7(a) | |
| | Incorrect Planning | The gold SQL involves nested queries, intermediate result processing through CTEs, or set operations for merging sub-query results. The model either fails to recognize or misuses these elements. | Refer to Fig. 7(b) | |
| Join Error | | The JOIN condition selects the incorrect tables or columns. | SELECT ... FROM spider2-public-data.wide_world_importers.sales_Invoices T1 JOIN spider2-public-data.wide_world_importers.sales_InvoiceLines T2 ON T1.OrderID = T2.InvoiceID | SELECT ... FROM `spider2-public-data.wide_world_importers.sales_InvoiceLines` AS INVL INNER JOIN `spider2-public-data.wide_world_importers.sales_Invoices` AS INV ON INVL.InvoiceID = INV.InvoiceID |
| Condition Filter Error | | The filtering conditions in the WHERE clause are incorrectly defined. | WHERE code.coding[OFFSET(0)].code IN ('44054006', '38341003') | WHERE code.coding[safe_offset(0)].display = 'Diabetes' OR code.coding[safe_offset(0)].display = 'Hypertension' |
| Misunderstanding External Knowledge | | The model misinterprets external knowledge relevant to the question. | Refer to Fig. 25 | |
| Excessive Prompt Length | | The input prompt exceeds the LLM's maximum length, causing truncation and making the answer inaccessible. | - | |
| Syntax Error | | The generated SQL query contains invalid syntax that prevents execution. | - | |

Figure 22: Descriptions and examples for all error categories.

## C.4 OTHER ANALYSIS

**Current LLM-based methods exhibit limited capability in addressing enterprise text-to-SQL tasks.** Tab. 5 illustrates that Spider 2.0-lite and Spider 2.0-snow present significant challenges. The highest performing method, DAIL-SQL + GPT-4o, achieves an EX of only 5.68%, which is markedly lower compared to its score of 86.6% on Spider 1.0 and 57.4% on BIRD datasets. With efficiently filtering the minimal sufficient schema, CHESS + GPT-4o is able to tackle more instances than DIN-SQL. Despite being extensively fine-tuned, SFT CodeS-15B is far from solving Spider 2.0-lite, with an EX score of only 0.73%, which further reveals the significant complexity gap between Spider 2.0-lite and the current text-to-SQL corpus. For Spider 2.0-snow, even the best method achieves only 2.20% EX, highlighting the increased challenge due to SQL dialect differences.

**LLM-agent frameworks face challenges handling different SQL dialects.** As shown in Tab. 20, we analyze the examples in Spider 2.0 based on database type. The results indicate that examples of the Snowflake type in Spider 2.0 are the most challenging. To assess the impact of SQL dialects on performance, we randomly select 180 examples and hosted them on both BigQuery and Snowflake, using the same set of questions. We find that performance on BigQuery is 12.78%, while on Snowflake it is 6.6%, highlighting that subtle differences in SQL dialect syntax can lead to variations in SQL generation performance.

## C.5 EXPERIMENTS COST

We summarize the average cost of API calls for each instance across different methods in Tab. 21.

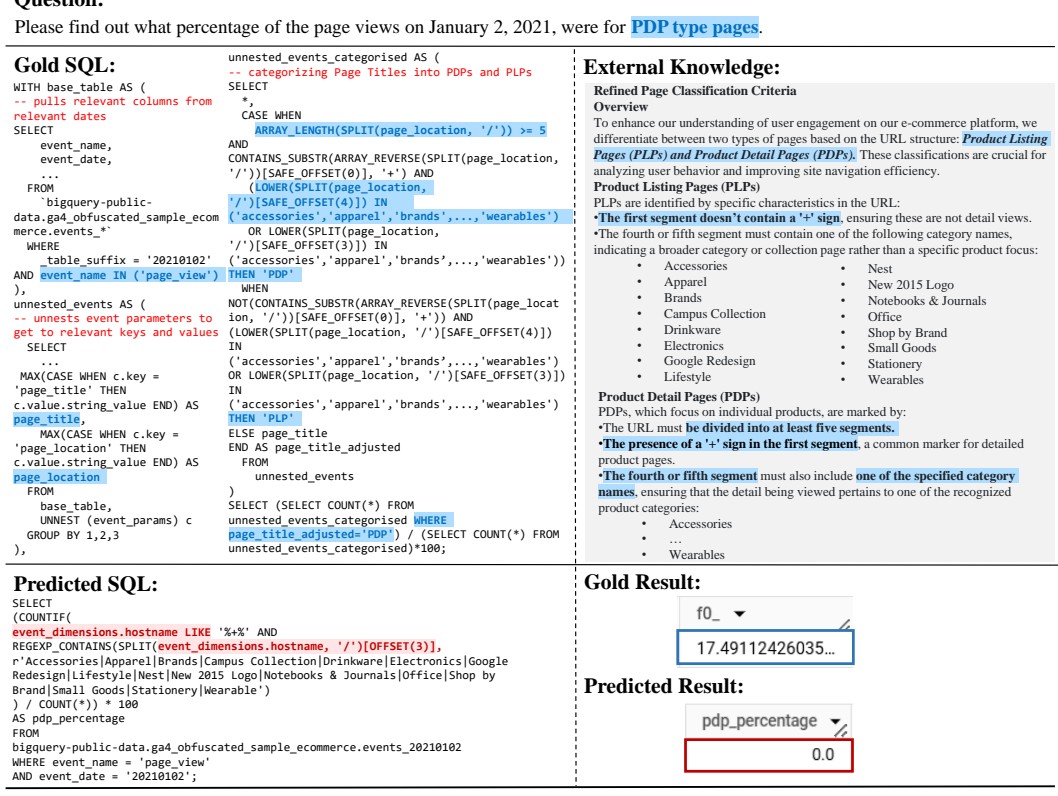

Figure 23: An example of **Misunderstanding External Knowledge**. The error in predicted SQL stems from the failure to correctly interpret the external knowledge provided for classifying `PDP` and `PLP` pages. While the predicted SQL uses a simple pattern-matching approach with regular expressions, it overlooks key aspects of the classification rules, such as the specific URL structure and the position of a `'+'` sign, which indicates the misunderstanding when trying to leverage the external knowledge.

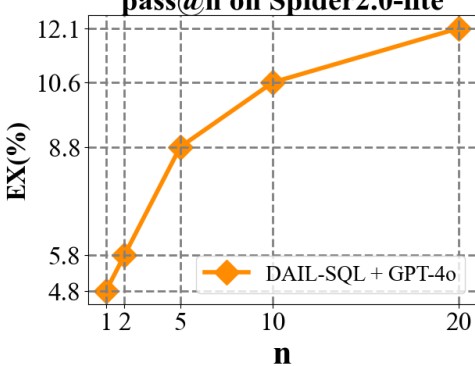

Figure 24: pass@{n} results.

Table 20: Model performance with different DB types.

| Task Subset | #Examples | SR (↑) |
| --- | --- | --- |
| Spider 2.0 | 100.00% | 17.0% |
| - w/ Bigquery | 33.86% | 24.07% |
| - w/ Snowflake | 31.33% | 7.14% |
| - w/ SQLite | 21.36% | 20.74% |
| - w/ DuckDB | 10.76% | 17.69% |
| - w/ Postgres | 1.58% | 12.82% |
| - w/ Clickhouse | 1.11% | 57.14% |

Table 21: Average cost per instance across all methods.

| Method | Avg. Cost (↓) |
|---|---|
| Spider-Agent + o1-preview | 0.75 $ |
| Spider-Agent + GPT-4-Turbo | 0.58 $ |
| Spider-Agent + GPT-4o | 0.32 $ |
| DIN-SQL + GPT-4o | 0.14 $ |
| DAIL-SQL + GPT-4o | 0.09 $ |
| DAIL-SQL + o1-preview | 0.32 $ |
| CHESS + GPT-4o | 0.43 $ |
| SFT CodeS-15B | 0.00 $ |

## C.6    CASE STUDY OF SPIDER-AGENT

In this section, we present the success and fail cases on Spider 2.0 using the o1-preview and Spider-Agent framework.

Fig. 25 shows a success case of google patents data. The Spider-Agent receives feedback from executed queries, allowing it to iteratively refine SQL queries until it achieves a bug-free result. Automated iterative debugging is one of the key advantages of Spider-Agent.

Fig. 26 shows a success case where Spider-Agent successfully solves a DBT project example. It first explores the codebase and database, locates where the code needs to be added, writes two SQL scripts, debugs, and runs the DBT command, ultimately completing the data transformation successfully.

Fig. 27 shows a failure case where the agent fails to explore the codebase and locate key information about monthly recurring revenue, resulting in an incorrect SQL query.

Fig. 28 presents a failure case where the model reads the document but fails to transform the complex document strings into an accurate SQL query.

Fig. 29 illustrates a failure where the model cannot comprehend the database format with nested columns, leading to multiple unsuccessful attempts at writing the correct code.

---

**Success case 1: about patents analysis**

### 17 Steps

**Action 0**: Bash(code="ls")
**Action 1**: Bash(code="cat README.md")
**Action 2**: BQ_GET_TABLES(database_name="patents-public-data", dataset_name="patents", save_path="dataset_metadata.csv")
**Action 3**: BQ_GET_TABLE_INFO(database_name="patents-public-data", dataset_name="patents", table="publications", save_path="publication_info.csv")
**Action 4**: BQ_EXEC_SQL(sql_query=..., is_save=False)
**Observation-4**: Error occurred while fetching data: 400 No matching signature for function DATE Argument types: INT64 Signature: DATE(TIMESTAMP, [STRING]) Argument 1: Unable to coerce type INT64 to expected type TIMESTAMP
**Thought-4**: The error indicates that I'm trying to use the `DATE` function on an `INT64` field, which is invalid. The date fields in the dataset are stored as integers in `YYYYMMDD` format. I need to convert these integer date fields to `DATE` using the `PARSE_DATE` function. I'll update my SQL query to properly parse the dates and include the condition for published applications.
......
**Action-14**: BQ_EXEC_SQL(sql_query=..., is_save="/workspace/result.csv")
**Observation-14**: Error occurred while fetching data: 400 No matching signature for function DATE Argument types: INT64 Signature: DATE(TIMESTAMP, [STRING]) Argument 1: Unable to coerce type INT64 to expected type TIMESTAMP
**Action-15**: BQ_EXEC_SQL(sql_query=..., is_save="/workspace/result.csv")
**Observation-15**: Now that the results are saved to `/workspace/result.csv`.
**Action-16**: Terminate(output="result.csv")

Figure 25: Success case 1. After multiple rounds of SQL writing and repeated debugging, the correct answer was obtained.

---

**Success case 2: DBT data transformation task**

### 20 Steps

......
**Action 3**: Bash(code="ls models/salesforce")
**Action 4**: Bash(code="cat models/salesforce/salesforce.yml")
**Action 5**: Bash(code="cat models/salesforce/salesforce_manager_performance.sql")
**Action 6**: Bash(code="ls models/salesforce/intermediate")
**Action 7**: Bash(code="cat models/salesforce/intermedi/int_salesforce__opportunity_by_owner.sql")
**Action 8**: LOCAL_DB_SQL(file_path="salesforce.duckdb", command="SHOW TABLES;", output="direct")
**Action 9**: LOCAL_DB_SQL(file_path="salesforce.duckdb", command="SHOW TABLES;", output="direct")
**Action 10**: CreateFile(filepath=models/salesforce/salesforce_daily_sales_activity.sql': with date_spine as ( select date_day as activity ... )
**Action 11**: Bash(code="dbt run")
**Observation 11**: I noticed warnings during 'dbt run' indicating that 'salesforce__daily_activity' is not found. Since we don't have a model named 'salesforce__daily_activity', I should remove references to it from 'salesforce.yml'. Additionally, I need to add the new model 'salesforce__daily_sales_activity' to 'salesforce.yml' with appropriate descriptions
......
**Action 15**: LOCAL_DB_SQL(file_path="salesforce.duckdb", command="SELECT * FROM salesforce__daily_sales_activity LIMIT 5;", output="direct")
......
**Action 18**: Bash(code="dbt run")
**Action 19**: Terminate(output="salesforce.duckdb")

Figure 26: Success Case 2.After multiple rounds of file and DBT operations, the data transformation task has been successfully resolved.

---

**Failure case 1**

**Action 1**: Bash(code="ls -R")
**Action 2**: Bash(code="cat models/customer_churn_month.yml")
**Action 3**: Bash(code="cat models/customer_churn_month.sql")
**Action 4**: EditFile(filepath="models/customer_churn_month.sql"):
. . . . . .
**Action 5**:Bash(code="cat models/mrr.yml")
**Action 6**:Bash(code="cat models/mrr.sql")
**Action 7**:EditFile(filepath="models/mrr.sql"):

```sql
with unioned as (
    select * from {{ ref('customer_revenue_by_month') }}
    union all
    select * from {{ ref('customer_churn_month') }}
),

lagged_values as (
    select
        *,
        coalesce(
            lag(is_active) over (partition by customer_id order by date_month), false
        ) as previous_month_is_active,
        coalesce( lag(mrr) over (partition by customer_id order by date_month), 0
        ) as previous_month_mrr
    from unioned
)

select
    date_month, customer_id, mrr, is_active,
    case
        when previous_month_is_active = false and is_active = true then 'New_Business'
        when previous_month_is_active = true and is_active = true and mrr_change > 0 then 'Expansion'
        when previous_month_is_active = true and is_active = true and mrr_change < 0 then 'Contraction'
        when previous_month_is_active = true and is_active = false then 'Churn'
        when previous_month_is_active = true and is_active = true and mrr_change = 0 then 'Renewal'
        when previous_month_is_active = false and is_active = false then 'Resurrection'
        else 'Other'
    end as change_category
from lagged_values
```

**Action 8**: Bash(code="dbt run")
**Action 9**: LOCAL_DB_SQL(file_path="mrr.duckdb", command="SELECT * FROM mrr LIMIT 5", output="direct")

Figure 27: Failure case 1. Reasonable data transformation process, but did not follow our predefined category type.

**Failure case 2**

**Action 1**: Bash(code="ls -R")
**Action 2**: Bash(code="cat README.md ga4_page_category.md")
**Action 3**: BQ_GET_TABLES(database_name="bigquery-public-data",
dataset_name="ga4_obfuscated_sample_ecommerce",save_path="dataset_metadata.csv")
**Action 4**: Bash(code="cat dataset_metadata.csv")
**Action 5**: BQ_GET_TABLE_INFO(database_name="bigquery-public-data",dataset_name=
"ga4_obfuscated_sample_ecommerce", table="events_20210109", save_path="ga4_info.csv")

```sql
WITH pdp_pages AS (
  SELECT
    event_params.value.string_value AS page_location
  FROM
    `bigquery-public-data.ga4_obfuscated_sample_ecommerce.events_20210102`,
    UNNEST(event_params) AS event_params
  WHERE
    event_name = 'page_view'
    AND event_params.key = 'page_location'
    AND REGEXP_CONTAINS(event_params.value.string_value, r'
        \/[^\/]+\/[^\/]+\/[^\/]+\/[^\/]+\/[^\/]*\+[^\/]*\/')
    AND REGEXP_CONTAINS(event_params.value.string_value, r'\/(Accessories|Apparel|Brands|Campus_Collection|
        Drinkware|Electronics|Google_Redesign|Lifestyle|Nest|New_2015_Logo|Notebooks_&_Journals|Office|
        Shop_by_Brand|Small_Goods|Stationery|Wearables)\/')
),
total_page_views AS (
  SELECT
    COUNT(*) AS total_views
  FROM
    `bigquery-public-data.ga4_obfuscated_sample_ecommerce.events_20210102`
  WHERE
    event_name = 'page_view'
)
SELECT
  (SELECT COUNT(*) FROM pdp_pages) * 100.0 / total_page_views.total_views AS pdp_percentage
FROM
  total_page_views
```

......
**Action 9**: Terminate(output="result.csv")

Figure 28: Failure case 2. Cannot ground the complex document into SQL.

**Failure case 3**

**Action 1**: Bash(code="ls -R")
**Action 2**: BQ_GET_TABLES(database_name="bigquery-public-data",
dataset_name="ga4_obfuscated_sample_ecommerce",save_path="dataset_metadata.csv")
**Action 3**: Bash(code="cat dataset_metadata.csv")
**Action 4**: BQ_GET_TABLE_INFO(database_name="bigquery-public-data",dataset_name=
"ga4_obfuscated_sample_ecommerce", table="events_20210109", save_path="ga4_info.csv")

```sql
SELECT
  event_timestamp,
  event_params.key,
  event_params.value.string_value AS page_location
FROM
  `bigquery-public-data.ga4_obfuscated_sample_ecommerce.events_20210128`,
  UNNEST(event_params) AS event_params
WHERE
  user_id = '1362228'
  AND event_date = '20210128'
ORDER BY
  event_timestamp
```

......
**Action 9**: Terminate(output="result.csv")

Figure 29: Failure case 3. Cannot understand the complex nested schema.

## C.7 PROMPTS OF SPIDER-AGENT

Following the ReAct (Yao et al., 2022) prompting framework, which guides models in breaking down complex problems into manageable steps through an iterative chain of reasoning (Reasoning), action (Action), and observation (Observation) phases, where the current state is observed after each action before proceeding to the next step of reasoning, we build the Spider-Agent. Below, we will introduce the system messages and actions used within this framework.

**System Prompt**

---

**System Prompt**

You are a data scientist proficient in database, SQL and DBT Project. You are starting in the **{work_dir}** directory, which contains all the data needed for your tasks. You can only use the actions provided in the **action space** to solve the task.

### [Action space]: **{action prompts}**

### [Notice]
1. First, run "ls' to check the current folder for files. If there are other markdown files, read them as they may contain useful information.
2. Examine the database schema folder, you fully understand the structure schema of the database.
3. Use appropriate SQL execution action to run queries.
4. Be prepared to write multiple SQL queries to find the correct answer. If an error occurs, revisit the database information and previous queries to adjust your SQL accordingly.
5. Ensure the results are valid. If the result.csv file is empty or only contains a header, the SQL query is incorrect. The final result should be either saved as a CSV or directly provided as a text answer, not an intermediate step or SQL statement.
6. After completing the task, verify the output data against the definitions. For dbt projects, after writing the SQL, run dbt run to update the database and confirm the new data models meet the YAML file definitions.

### [Response format]
For each task input, your response should contain:
1. One analysis of the task and the current environment, reasoning to determine the next action (prefix "Thought: ").
2. One action string in the ACTION SPACE (prefix "Action: ").

### [Example interaction]
Observation: ...(the output of last actions, as provided by the environment and the code output, you don't need to generate it)
Thought: ...
Action: ...

### [Task]: **{Task}**

---

**Action Space Prompt**

---

**Bash**

## Bash Action
* Signature: Bash(code="shell_command")
* Description: This action string will execute a valid shell command in the code field. Only non-interactive commands are supported. Commands like "vim" and viewing images directly (e.g., using "display") are not allowed.
* Example: Bash(code="ls -l")

---

**CreateFile**

## CreateFile Action
* Signature: CreateFile(code="shell_command')
* Description: This action string will execute a valid shell command in the 'code' field. Only non-interactive commands are supported. Commands like "vim' and viewing images directly (e.g., using "display') are not allowed.
* Example: CreateFile(code="ls -l")

---

**EditFile**

## EditFile
* Signature: EditFile(filepath="path/to/file"):
```
File_content
```
* Description: This action will overwrite the file specified in the filepath field with the content wrapped in paired symbols. Normally, you need to read the file before deciding to use EditFile to modify it.
* Example: EditFile(filepath="hello_world.py"):
```
print("Hello, world!")
```

**BIGQUERY_EXEC_SQL**

## BIGQUERY_EXEC_SQL
* Signature: BIGQUERY_EXEC_SQL(sql_query="SELECT * FROM your_table", is_save=True, save_path="/workspace/output_file.csv")
* Description: Executes a SQL query on Google Cloud BigQuery. If "is_save' is True, the results are saved to a specified CSV file; otherwise, results are printed. If you estimate that the number of returned rows is small, you can set is_save=False, to directly view the results. If you estimate that the number of returned rows is large, be sure to set is_save = True. The 'save_path' CSV must be under the '/workspace' directory.
* Examples:
- Example1: BIGQUERY_EXEC_SQL(sql_query="SELECT count(*) FROM sales", is_save=False)
- Example2: BIGQUERY_EXEC_SQL(sql_query="SELECT user_id, sum(purchases) FROM transactions GROUP BY user_id", is_save=True, save_path="/workspace/result.csv")

**GET_TABLES**

## GET_TABLES
* Signature: GET_TABLES(database_name="your_database_name", dataset_name="your_dataset_name", save_path="path/to/output_file.csv")
* Description: Executes a query to fetch all table names and their corresponding DDL from the specified dataset in Google Cloud BigQuery. The results are saved to the specified CSV file.
- The BigQuery id of a table is usually in the form of database_name.dataset_name.table_name. This action mainly focuses on the tables under dataset_name.
* Examples:
- Example1: GET_TABLES(database_name="bigquery-public-data", dataset_name="new_york", save_path="dataset_metadata.csv")

**GET_TABLES_INFO**

## GET_TABLE_INFO Action
* Signature:
GET_TABLE_INFO(database_name="your_database_name", dataset_name="your_dataset_name", table="table_name", save_path="path/to/output_file.csv")
* Description: Executes a query to fetch all column information (field path, data type, and description) from the specified table in the dataset in Google Cloud BigQuery. The results are saved to the specified CSV file.
- The BigQuery id of a table is usually in the form of database_name.dataset_name.table_name.
* Examples:
- Example1: GET_TABLE_INFO(database_name="bigquery-public-data", dataset_name="samples", table="shakespeare", save_path="shakespeare_info.csv")

**SAMPLE_ROWS**

## SAMPLE_ROWS Action
* Signature:
SAMPLE_ROWS(database_name="your_database_name", dataset_name="your_dataset_name", table="table_name", save_path="path/to/output_file.csv")
* Description: Executes a query to fetch all column information (field path, data type, and description) from the specified table in the dataset in Google Cloud BigQuery. The results are saved to the specified CSV file.
- The BigQuery id of a table is usually in the form of database_name.dataset_name.table_name.
* Examples:

- Example1: SAMPLE_ROWS(database_name="bigquery-public-data", dataset_name="samples", table="shakespeare", save_path="shakespeare_info.csv")

