# OpenReview forum: "Spider 2.0: Evaluating Language Models on Real-World Enterprise Text-to-SQL Workflows"
_ICLR.cc/2025/Conference — ICLR 2025 Oral_

### Official Review · Reviewer_c1AS · 2024-10-28

**Soundness:** 3
**Presentation:** 4
**Contribution:** 4
**Rating:** 8
**Confidence:** 5

**Summary:**

The paper presents Spider 2.0, a new benchmark for Text-to-SQL systems that introduces more complex and realistic SQL tasks compared to benchmarks like Spider 1.0 and BIRD. Spider 2.0 tasks require models to leverage project-level information, such as codebases and documentation, for generating accurate SQL queries. It also includes more SQL dialects such as DuckDB, Snowflake, and BigQuery, and utilizes advanced SQL functions such as ST_DISTANCE and CORR.

The benchmark demonstrates its difficulty by showing that even state-of-the-art language models achieve a maximum accuracy of only 15.12%. This result highlights significant gaps in current Text-to-SQL approaches that perform well on enterprise-level SQL generation tasks. Spider 2.0 can potentially advance Text-to-SQL systems, bringing them closer to being practical for enterprise applications.

**Strengths:**

1. Spider 2.0 is the first Text-to-SQL benchmark to incorporate complex contexts — such as data pipeline code and SQL dialect documentation — into the SQL generation process, reflecting an essential aspect of real-world SQL writing in a data engineer’s workflow. This marks a significant advancement over BIRD, which only includes simple evidence (1-2 sentences) as external knowledge.
2. SQL queries in Spider 2.0 are derived from actual projects and tutorials, enhancing the benchmark's realism and quality. Each database instance is selected to ensure sufficient complexity, with a minimum of 200 columns per database.
3. The evaluation in this paper is highly insightful, offering readers a clear perspective on the current limitations of LLM-based text-to-SQL approaches.

**Weaknesses:**

The goal of this paper is to simulate real-world enterprise text-to-SQL workflows. While the authors have made substantial progress toward this aim, several important aspects of enterprise workflow characteristics remain unaddressed.

1. Context Complexity: The paper has included codebase and SQL docs as the context. However, in the enterprise setting, business context is also crucial. For example, different companies have different ways to compute daily active users or user retention rates. There can be a page-long documentation to explain how a metric should be calculated or even a page-long formula that defines a metric directly. This kind of context is missing from the benchmark.
2. Schema Complexity: The paper says it selects databases where the number of columns is larger than 200. However, that does not guarantee that the schema is complex enough. For example, these 200 columns can be stored in one table. It is not clear how complicated the table relationship is. How many joins are involved in the SQL?
3. Question Complexity: The questions are generated by the authors based on the SQLs crawled from forums or tutorials, which might not reflect the nature of real-world BI questions, which often ask for trends or patterns. Including real data engineers to audit the benchmark or participate in the benchmark generation process can make it more convincing.

Despite these limitations, I recognize the authors' efforts in taking an important first step toward an enterprise setting.

**Questions:**

1. How many tables per database in Spider 2.0? The paper only reports the number of columns per database.
2. The paper's abstract states that it can "generate multiple SQL queries with diverse operations, often exceeding 100 lines." However, the queries average 144 tokens, which seems unlikely to translate into over 100 lines. Am I misunderstanding something here?

---

> ### Author Response · Authors · 2024-11-21
> **Authors' Response [1/2]**
>
> Thank you for your suggestion regarding the characteristics of enterprise workflows. We will provide clarification on this matter.
>
> **Q1: Context Complexity**
>
> A1: Spider 2.0 includes many documents as context, providing detailed descriptions of SQL writing specifics, metric calculations, and more. We believe these documents meet enterprise-level standards.
>
> For example, as shown in Figures 14 and 15, we present the dimension and metrics analysis documents from Google Analytics (`ga4_dimensions_and_metrics.md`), which provide detailed explanations of data analytics processes. The figures display only a portion of the content, while the actual document is much longer. Additionally, there are other extensive documents, such as Mitelman's document on correlation calculations and `ethereum_data_transformation`. Using tiktoken package for tokenization, we found that the average document length is 2,037 tokens. In the case of DBT projects, there are even more documents to include.
>
> **Q2: Schema Complexity**
>
> A2: The databases in Spider 2.0 involve a large number of tables and columns. Due to space limitations, we provided statistics in Appendix B.8, Figure 19, showing that each database contains an average of 52 tables, and each SQL query requires **joins** across an average of 4.8 tables. Each database schema (tokenizing only the DDL statements) averages 20,478 tokens. If column descriptions are included, the size becomes significantly larger. This highlights the complexity of Spider 2.0's schema.
>
>
> **Q3: Question Complexity**
>
> A3: Thank you for your insightful discussion. This is an issue we considered during the data collection process. In the first round of data collection, we identified 1,021 queries. Subsequently, we performed data filtering, with one of the key criteria being that the data must represent a real BI or data analysis problem. As a result, we removed some queries that did not meet the requirements, ultimately retaining approximately 540 examples.
> Although this approach may introduce a small amount of partially unnatural tasks, collecting questions from professional data warehouses such as BigQuery and Snowflake, as well as specialized platforms like Google Analytics and Salesforce, ensures that the data is relatively realistic.
>
>
> **Q4: How many tables per database in Spider 2.0?**
>
> A4: Due to space limitations, we provided statistics in Appendix B.8, Figure 19, showing that each database contains an average of 52 tables.

---

> ### Author Response · Authors · 2024-11-21
> **Authors' Response [2/2]**
>
> **Q5: The number of lines per SQL query**
>
> A5: Thank you for your thorough reading of our paper.
>
> When counting the number of lines, we adhere to SQL formatting conventions. For example, each selected column, each table in the FROM clause with a JOIN, and each condition are counted as separate lines. Additionally, each column or table is treated as an individual token during token count calculations, as we use whitespace as the delimiter.
>
> As a result, the number of lines in SQL queries **often** exceeds 100, although Spider 2.0 also includes SQL queries with fewer than 100 lines. Regardless of their length, the complexity of the SQL queries remains high.
> Here is a long SQL example.
>
> ```
> WITH fam AS (
>   SELECT
>     DISTINCT family_id
>   FROM
>     `spider2-public-data.patents_google.publications`
> ),
>
> crossover AS (
>   SELECT
>     publication_number,
>     family_id
>   FROM
>     `spider2-public-data.patents_google.publications`
> ),
>
> pub AS (
>   SELECT
>     family_id,
>     MIN(publication_date) AS publication_date,
>     STRING_AGG(DISTINCT(p.publication_number) ORDER BY p.publication_number) AS publication_number,
>     STRING_AGG(DISTINCT(country_code) ORDER BY country_code) AS country_code
>   FROM
>     `spider2-public-data.patents_google.publications` AS p
>   GROUP BY
>     family_id
> ),
>
> tech_class AS (
>   SELECT
>     family_id,
>     STRING_AGG(DISTINCT(cpc.code) ORDER BY cpc.code) AS cpc,
>     STRING_AGG(DISTINCT(ipc.code) ORDER BY ipc.code) AS ipc
>   FROM
>     `spider2-public-data.patents_google.publications` AS p,
>     UNNEST(cpc) AS cpc,
>     UNNEST(ipc) AS ipc
>   GROUP BY
>     family_id
> ),
>
> cit AS (
>   SELECT
>       p.family_id,
>       STRING_AGG(DISTINCT(crossover.family_id)) AS citation
>     FROM
>       `spider2-public-data.patents_google.publications` AS p,
>       UNNEST(citation) AS citation
>       LEFT JOIN
>         crossover
>         ON citation.publication_number = crossover.publication_number
>     GROUP BY
>       p.family_id
> ),
>
> tmp_gpr AS (
>       SELECT
>         family_id,
>         SPLIT(STRING_AGG(DISTINCT(cited_by.publication_number))) AS cited_by_publication_number
>       FROM
>         `spider2-public-data.patents_google.abs_and_emb` AS p,
>         UNNEST(cited_by) AS cited_by
>       LEFT JOIN
>         crossover
>       ON
>         p.publication_number = crossover.publication_number
>       GROUP BY
>         family_id
> ),
>
> gpr AS (
>       SELECT
>         tmp_gpr.family_id,
>         STRING_AGG(DISTINCT(crossover.family_id)) AS cited_by
>     FROM
>       tmp_gpr,
>       UNNEST(cited_by_publication_number) AS cited_by_publication_number
>     LEFT JOIN
>       crossover
>     ON
>       cited_by_publication_number = crossover.publication_number
>     GROUP BY
>       tmp_gpr.family_id
> )
>
> SELECT
>   fam.family_id,
>   pub.* EXCEPT(family_id),
>   tech_class.* EXCEPT(family_id),
>   cit.* EXCEPT(family_id),
>   gpr.* EXCEPT(family_id)
> FROM
>   fam
> LEFT JOIN
>   pub
> ON
>   fam.family_id = pub.family_id
> LEFT JOIN
>   tech_class
> ON
>   fam.family_id = tech_class.family_id
> LEFT JOIN
>   cit
> ON
>   fam.family_id = cit.family_id
> LEFT JOIN
>   gpr
> ON
>   fam.family_id = gpr.family_id
> WHERE
>   publication_date BETWEEN 20150101 AND 20150131
>
> ```
>
>
> However, the number of lines is not a standardized metric.
>
> Therefore, in the main text, we use tokens as the unit of measurement because token count serves as a consistent metric across all SQL datasets, whereas line count can vary depending on formatting styles.
>
> That said, in terms of SQL complexity, Spider 2.0 significantly surpasses existing popular datasets. For example, Spider 2.0-lite has an average token count of 144.5, while BIRD averages 30.9.
>
> ---
>
> Finally, thank you for your thoughts and valuable feedback! We hope we have addressed most of your concerns. If you have any questions, please don't hesitate to reach out to us.

---

> ### Author Response · Authors · 2024-11-27
> **Follow-up**
>
> Dear Reviewer,
>
> We'd appreciate it if you'd let us know if our response has addressed your concerns.
>
> Thanks!

---

> > ### Comment · Reviewer_c1AS · 2024-11-28
> >
> > Thank authors for the detailed clarification. It addresses my questions and concerns!

---

### Official Review · Reviewer_Bdrh · 2024-11-04

**Soundness:** 4
**Presentation:** 4
**Contribution:** 4
**Rating:** 8
**Confidence:** 4

**Summary:**

This paper introduces Spider 2.0, a benchmark aimed at assessing the capabilities of large language models (LLMs) in handling complex text-to-SQL tasks within enterprise environments. Unlike previous datasets, Spider 2.0 involves real-world databases containing thousands of columns and diverse SQL dialects, with tasks that require interaction with database metadata, project-level codebases, and external documentation. The benchmark emphasizes challenges such as processing extensive schemas, using advanced SQL functions, and generating multi-step SQL queries.

Key findings highlight that current LLMs, even state-of-the-art models like GPT-4, struggle with Spider 2.0, achieving a success rate of only 15.1%. This reveals significant gaps in their ability to handle industrial-grade SQL workflows, which involve complex data transformations and analytical tasks. Spider 2.0 aims to push the development of more capable models for practical enterprise applications by providing a realistic and challenging benchmark.

**Strengths:**

S1. It is important to have Spider 2.0 for real enterprise Text-to-SQL. Spider 1.0 and BIRD focuses on multi-table tasks, while WikiSQL and GeoQuery only focus on single table tasks. These datasets are popular, but they still represent a very small part of the whole data analytics/science workflow. The emerge of Spider 2.0 tries to fill this gap.

S2. The collection of Spider 2.0 complies the data manipulation workflow in real enterprise scenarios. Detailed types of queries and error analysis are shown in detail, which indicates the future direction of improvement.

S3. Spider 2.0 is a valuable dataset for the future agent of data analytics, which will be an important future trend of the data science domain. This paper also provides Spider-Agent, which can be a new baseline of data analytics agent.

**Weaknesses:**

W1. There should be an opportunity to have an end-to-end benchmark for enterprise data analytics/ data science workflow with both SQL query and Python. Also, it would be better to also consider queries on Hadoop/Cassandra, which are also widely-used storages.

W2. In this paper, schema linking and table mapping results are shown. It would be better to open-source these two related datasets to extend the gap of LLM-based Text-to-SQL schema linking.

W3. It would be better to show the number of tokens of Codebase+DB Info+Documents, which should be an important parameter for picking up proper LLMs to construct agents.

**Questions:**

1. W3

2. May I ask where can I find the proper json file for Spider 2.0 and Spider 2.0 Light in your repo?

3. Please figure out the difference between your work and MLE-Bench made by OpenAI

---

> ### Author Response · Authors · 2024-11-21
> **Authors' Response**
>
> Thank you for your insightful review and for recognizing the novelty of our work.
>
> **Q1: End-to-end benchmark for enterprise data analytics/ data science workflow with both SQL query and Python**
>
> **A1**: Thank you for your suggestion. Spider 2.0, as a comprehensive task, enables Agent methods to perform holistic data analytics tasks by combining the use of SQL and Python.
>
> According to statistics, in the Spider-Agent + o1-preview experiments, the agent utilized Pandas for post-processing in 14.3% of the examples. SQL offers higher execution efficiency, while Python+Pandas enables more convenient handling of intermediate results from SQL execution. Exploring how to combine them efficiently is a promising direction for future research.
>
> Spider 2.0 centers on SQL generation, encompassing tasks involved in data engineering such as data wrangling, data transformation, and data analytics. This is currently the most realistic dataset in the field of data engineering. In the future, with the development of LLMs, we believe the community can create even more realistic, end-to-end datasets for enterprise data analytics workflows.
>
>
>
> **Q2: Hadoop/Cassandra**
>
> **A2**: This is an excellent suggestion, which might require the development of a more comprehensive data engineering benchmark involving a wider range of software technologies. This paper focuses specifically on the text-to-SQL aspect of data engineering.
>
> **Q3: Open-source schema linking and table mapping results**
>
> **A3**: Thank you for your suggestion. Open-sourcing the linking and table mapping results will indeed be valuable for the entire community. We will release the gold tables and columns for each instance in a single JSON file.
>
> **Q4: Show the number of tokens of Codebase+DB Info+Documents**
>
> **A4**: Spider 2.0 is an agentic code generation task, where it is usually unnecessary to input the entire context, including the codebase, database information, and documents, into the LLM at once. We do not recommend using this approach to handle the Spider 2.0 dataset. However, we believe your suggestion is meaningful. We used the tiktoken package to tokenize the context in Spider 2.0. The average document length is 2,037 tokens, while each database schema (tokenizing only the DDL statements) averages 20,478 tokens. If column descriptions are included, the size becomes significantly larger. Based on your suggestion, we will provide an additional metadata JSON file that specifies the maximum number of context tokens required for each task.
>
>
>
> **Q5: The difference between Spider 2.0 and MLE-Bench.**
>
> **A5**: Although Spider 2.0 and MLE-Bench are both complex code generation tasks, they focus on distinct areas. MLE-Bench primarily targets machine learning tasks, while Spider 2.0 is centered on data engineering, particularly text-to-SQL tasks. Unlike MLE-Bench, Spider 2.0 does not involve ML/DL model training but instead emphasizes enterprise-level text-to-SQL tasks such as data wrangling, data transformation, and data analytics. The domains covered by these two benchmarks are orthogonal, and we believe they can together drive advancements in automated data science and engineering.
>
>
>
> **Q6: May I ask where can I find the proper json file for Spider 2.0 and Spider 2.0 Light in your repo?**
>
> You can find the respective files at `spider2-lite/spider2-lite.json` for Spider 2.0-lite and `spider2/examples/spider2.jsonl` for Spider 2.0 in our repository.
>
> ---
>
> Finally, thank you for your thoughts and valuable feedback! We hope we have addressed most of your concerns. If you have any questions, please don't hesitate to reach out to us.

---

> ### Author Response · Authors · 2024-11-27
> **Follow-up**
>
> Dear Reviewer,
>
> We'd appreciate it if you'd let us know if our response has addressed your concerns.
>
> Thanks!

---

### Official Review · Reviewer_XhZP · 2024-11-05

**Soundness:** 3
**Presentation:** 3
**Contribution:** 4
**Rating:** 8
**Confidence:** 5

**Summary:**

The paper introduces a new benchmark, Spider 2.0, which consists of 595 real-world text-to-SQL workflow tasks over ~1000 columns. The  text-to-SQL tasks are significant more challenging compared to the commonly used Spider and Bird benchmarks. The authors evaluate multiple LLMs using Spider 2.0 and present a set of interesting findings.

**Strengths:**

S1. The new benchmark dataset is publicly available and critical for the community and enterprises that develop LLM-based text-to-SQL solutions.

S2. The dataset creation is solid and clearly presented in the paper. The SQL complexity including functions and dialects is closer to the real-life enterprise setting.

S3. The analysis provides some interesting findings, especially the error analysis of SQL generation.

**Weaknesses:**

W1. The agent aspect of the benchmark should be emphasized more as it largely separates Spider 2 from the other benchmarks. It is reasonable to use execution results as the metric for evaluation. However, the other important aspect is the ideal multi-turn process for each task. For example, what is the typical number of turns and the sequence to complete the task.

W2. The difficult level in Sec. 3.1 is acceptable but not comprehensive. The SQL complexity reflects one aspect of the task difficulty. The other aspects include the data complexity (e.g., whether the task requires access to nested schema, any question/data ambiguity, etc.). If these are not as straightforward as the number of tokens, associating some tags to each task would help one to understand the specific challenges associated with the tasks.

W3. It seems that SQL is the only "code" allowed to execute the task. However, certain tasks such as data transformation might be easier to be executed through script languages (e.g., Python). IMO, a more realistic setting should allow LLMs to decide the best way to execute a (sub-)task. This will enable the LLMs, such as GPT-4o, to better leverage their strength in code generation.

**Questions:**

Q1. Is CodeS-15B fine-tuned on Spider 2, or the fine-tuned model from the repo was directly used for evaluations?

Q2. Among dialect function usage errors, what is the ratio between simple syntax errors (i.e., LLMs not knowing the specific syntax) and completely incorrect chosen function?

---

> ### Author Response · Authors · 2024-11-21
> **Authors' Response**
>
> Thank you for your insightful review and for recognizing the novelty of our work.
>
> **Q1: The agent aspect of the benchmark should be emphasized more. What is the typical number of turns and the sequence to complete the task?**
>
> A1: Thank you for your insightful suggestion. The current metric is somewhat incomplete from a certain perspective. For an agent task, it is more meaningful to have diverse evaluation metrics.
>
> Number of turns is indeed a meaningful metric. According to our experimental analysis, a more advanced Agent framework with a complex action space, typically results in more turns. Our evaluation prioritizes problem-solving performance, so we use *successful rate* as the primary metric. However, Efficiency (Number of turns) remains an important auxiliary metric for future research. In Appendix C.1 and Figure 21, we quantitatively and qualitatively analyzed the number of action steps and action types, respectively.
>
> In future versions, we plan to include more auxiliary metrics to comprehensively evaluate the performance of code generation agents, such as the number of actions and time efficiency scores.
>
>
> **Q2: The difficulty level in Sec. 3.1 is acceptable but not comprehensive.**
>
> A2: **We agree with your suggestions**—the current classification of Easy/Medium/Hard is not comprehensive. However, incorporating many dimensions such as database complexity and SQL complexity to generate a composite difficulty score and identifying an optimal weighting method for these dimensions is challenging. Therefore, we used the most straightforward metric—SQL length—as an evaluation criterion.
>
> For other dimensions, we have conducted corresponding experimental analyses:
> - Number of DB columns: Figure 5
> - Number of Joins (table numbers): Figure 21
> - Nested columns: Table 8
> - External Document: Table 7
>
> In the future, based on your suggestion, we will also tag each example with its characteristics in a metadata JSON file, making it easier for participants to understand the examples.
>
> **Q3: It seems that SQL is the only "code" allowed to execute the task.**
>
> A3: There seems to be some misunderstanding here. The Spider 2.0 task setting actually allows Agents to use other programming languages, including Python. As shown in Figure 22, we have demonstrated the types of actions performed by the Spider-Agent to complete tasks. Additionally, based on the experimental logs, Spider-Agent often alternates between SQL queries and Python-Pandas when solving tasks, instead of relying exclusively on SQL.
>
> According to statistics, in the Spider-Agent + o1-preview experiments, the agent utilized Pandas for post-processing in 14.3% of the examples. SQL offers higher execution efficiency, while Python+Pandas enables more convenient handling of intermediate results from SQL execution. Exploring how to combine them efficiently is a promising direction for future research. Additionally, solving the tasks in Spider 2.0 requires writing bash scripts, a skill that Coding Agents must also possess.
>
> **Q4: Is CodeS-15B fine-tuned on Spider 2, or was the fine-tuned model from the repository directly used for evaluations?**
>
> A4: Since Spider 2.0 does not provide a training set, we directly used a fine-tuned model. This also underscores the challenge of Spider 2.0: the NLP community needs stronger and more complex text-to-SQL training corpora.
>
> **Q5: Among dialect function usage errors, what is the ratio between simple syntax errors (i.e., LLMs not knowing the specific syntax) and completely incorrect chosen functions?**
>
> A5: In dialect function usage errors, simple syntax errors account for 6.8% (which is similar to the proportion of "syntax errors" mentioned in Figure 6 (7.4%)), while completely incorrect chosen functions account for 93.2%.
>
> Btw, for the incorrect dialect function usage reported in Figure 6, some examples may also include syntax errors in addition to incorrectly chosen functions. Since we believe that incorrect dialect function usage represents a more insightful error category, we have classified these examples under this category.
>
> ---
>
> Finally, thank you for your thoughts and valuable feedback! We hope we have addressed most of your concerns. If you have any questions, please don't hesitate to reach out to us.

---

> ### Author Response · Authors · 2024-11-27
> **Follow-up**
>
> Dear Reviewer,
>
> We'd appreciate it if you'd let us know if our response has addressed your concerns.
>
> Thanks!

---

> > ### Comment · Reviewer_XhZP · 2024-11-28
> >
> > Thanks for the authors' responses. I have reviewed them along with comments from other authors. Most of my concerns have been addressed.

---

### Official Review · Reviewer_mvyZ · 2024-11-11

**Soundness:** 4
**Presentation:** 4
**Contribution:** 3
**Rating:** 8
**Confidence:** 4

**Summary:**

The paper introduces Spider 2.0, a new NL2SQL benchmark designed to address the challenges of real-world enterprise text-to-SQL workflows. Unlike previous benchmarks, Spider 2.0 focuses on 595 complex workflow problems derived from real-world enterprise use cases involving databases often exceeding 1,000 columns and stored across diverse systems like BigQuery and Snowflake. These problems require models to handle intricate SQL workflows, understand long contexts, explore database metadata, manage dialect variations, and interact with project-level codebases. The tasks frequently involve generating multi-step, nested SQL queries exceeding 100 lines, encompassing diverse operations from data transformation to analytics. Spider 2.0 thus pushes beyond traditional text-to-SQL challenges, demanding higher levels of reasoning and adaptability from language models.

The evaluations reveal significant gaps in existing language model capabilities, with the introduced code agent framework solving only 15.1% of Spider 2.0 tasks, compared to 91.2% on Spider 1.0 and 73.0% on BIRD. This highlights the difficulty of handling real-world enterprise scenarios, such as schema linking in large databases, navigating diverse SQL dialects, planning complex query sequences, and effectively leveraging external documentation. The insights from Spider 2.0 underscore the need for substantial advancements in autonomous code agents to meet enterprise requirements.

**Strengths:**

S1) Originality: Spide 12.0 significantly extends traditional text-to-SQL benchmarks by focusing on real-world enterprise scenarios.  It incorporates a broader and more complex range of challenges, such as handling extremely large databases, diverse SQL dialects, and intricate workflows. This originality lies in redefining the scope of text-to-SQL tasks to better align with practical, enterprise-level applications.

S2) Quality: the benchmark created and curated 595 real-world text-to-SQL workflow problems derived from enterprise use cases, ensuring relevance and rigor. The inclusion of diverse database systems like BigQuery and Snowflake, along with tasks involving database metadata, codebases, and nested SQL queries, reflects meticulous data collection and a comprehensive representation of real-world complexity.

S3) Clarity: the paper is very well organized and carefully written. The content is very rich and deep. By providing detailed analysis and explicit examples of schema linking, dialect handling, and multi-step query generation, it offers clear insights into the difficulties faced by language models in enterprise scenarios.

S4) Reproducibility: the paper provides source code.

**Weaknesses:**

I don't find any major weakness

**Questions:**

No questions

---

> ### Author Response · Authors · 2024-11-21
> **Authors' Response**
>
> Thank you for your thorough and detailed review of our paper, as well as your summary and recognition of our work. If you have any further questions, please feel free to let us know.

---

### Author Response · Authors · 2024-11-21
**Official Comment by Authors**

# To all reviewers

We would like to express our sincere gratitude to all the reviewers for their thoughtful and constructive feedback on our submission. We genuinely appreciate the time and effort invested in helping us refine our work.
We are also delighted that our benchmark is considered “*originality, quality, clarity, reproducibility*” (reviewer mvyZ), “*The dataset creation is solid*” and “*interesting findings, especially the error analysis*” (reviewer XhZP), “*Spider 2.0 is a valuable dataset for the future agent of data analytics*” (reviewer Bdrh) and “*The evaluation in this paper is highly insightful*” (reviewer c1AS).

We believe Spider 2.0 will make a highly impactful contribution. Its significance extends beyond the **text-to-SQL** domain: It builds upon the legacy of Spider 1.0 and BIRD, becoming the new most challenging text-to-SQL benchmark, while also serving as a general-purpose benchmark for **code generation and agent-based tasks**. Much like SWE-Bench and MLE-Bench, we envision Spider 2.0 as a foundational resource for advancing code intelligence research.

Our detailed responses address individual queries and suggestions from each reviewer, covering the following types of issues:

1. **More statistical information**: We have provided explanations for this information: difficulty level (Reviewer XhZP), context length (Reviewer Bdrh), number of tables (Reviewer c1AS) and SQL length (Reviewer c1AS).
2. **Complexity of task**: We addressed reviewer c1AS's concerns about complexity from three perspectives: Context, Schema, and Question.
3. **Evaluation metrics for agent task**: We appreciate Reviewer XhZP's suggestion and have provided an explanation of our existing analysis results on the agent evaluation metric.
4. **Can other programming languages be used**: Both Reviewer Bdrh and Reviewer XhZP raised concerns about whether Python can be used to solve the tasks in Spider 2.0. We have provided an explanation regarding this.
5. **Open-source more information**: As Reviewer Bdrh suggested, we will include more information, such as schema linking and table mapping details, when releasing Spider 2.0.


Once again, we thank all reviewers for their invaluable suggestions and insights, which have significantly contributed to the improvement of our paper. Please feel free to reach out if further clarification is required to assist in the final assessment.

---

> ### Author Response · Authors · 2024-12-02
> **Official Comment by Authors**
>
> Dear Reviewers,
>
> Thank you for your thorough and responsible review as well as your insightful suggestions. We also deeply appreciate your recognition of Spider 2.0!
>
> As the rebuttal period is coming to an end, please feel free to let us know if you have any additional suggestions or questions about Spider 2.0.
>
> Thank you once again for your valuable feedback.
>
> Best regards,
> Authors

---

### Public Comment · ~harshraj_bhoite1 · 2025-02-26
**Evaluating Spider 2.0: A Critical Review of Enterprise-Level Text-to-SQL Benchmarking**

Summary:
The paper presents Spider 2.0, a benchmark designed to evaluate the performance of language models in real-world enterprise text-to-SQL workflows. Unlike prior benchmarks, which often use simplified databases, Spider 2.0 includes diverse database systems (e.g., BigQuery, Snowflake, ClickHouse) and real-world queries that involve thousands of columns, complex SQL dialects, and project-level dependencies. The authors highlight that existing LLMs struggle significantly with these challenges, achieving only 15.1% success, compared to 91.2% on Spider 1.0. This finding underscores the need for more robust models and techniques to bridge the gap between research-focused text-to-SQL tasks and enterprise applications.

Soundness: 4 (Methodology is rigorous, but evaluation could be more flexible)
Presentation: 3 (Well-structured but could be more concise in places)
Contribution: 4 (Important benchmark for advancing text-to-SQL research)

Strengths:

Enterprise Relevance: Unlike prior text-to-SQL datasets, Spider 2.0 closely mirrors real-world enterprise scenarios, making it a highly valuable benchmark.

Comprehensive Benchmark: Covers multiple database types, SQL dialects, and data transformation tasks, making it far more realistic and challenging than previous datasets.

Dual Task Settings: Provides both Spider 2.0 (interactive agent setting) and Spider 2.0-lite (standard text-to-SQL conversion), allowing for a flexible evaluation framework.

Thorough Evaluation: The paper benchmarks multiple state-of-the-art models, highlighting clear performance gaps between research-focused and real-world SQL workflows.

Weaknesses:

Rigid Evaluation Approach: The success criteria assume a single correct SQL query, whereas enterprise SQL tasks often allow multiple valid solutions. Using execution-based evaluations would provide a more flexible performance measure.

Limited Human Evaluation Details: The study mentions manual annotation, but inter-annotator agreement, annotator expertise, and verification steps are not well-documented.

Scalability Concerns: The dataset features extremely large schemas (thousands of columns), but the paper does not explore schema filtering mechanisms that could help LLMs handle them more effectively.

Lack of Adaptive Strategies: While LLMs perform poorly, the study does not experiment with fine-tuning approaches or hybrid retrieval-augmented methods, which might offer significant improvements.

Questions:

How do you ensure fairness in evaluating models across different SQL dialects? Are there cases where models perform better on specific dialects due to training data biases?

What are the primary failure points in model-generated SQL? Are issues related to incorrect schema linking, JOIN errors, or SQL syntax mistakes?

Would this benchmark be extensible to SQL tasks that integrate with external BI tools or ETL workflows? This could make Spider 2.0 even more representative of enterprise SQL use cases.

Flag For Ethics Review: No ethics review needed.
Rating: 8 (Strong contribution, recommended for acceptance with minor revisions)
Confidence: 4 (Confident in the assessment, but some additional details could strengthen the conclusions)
Code Of Conduct: Yes

---

### Meta-Review · Area_Chair_HJY3 · 2024-12-22

**Metareview:**

The paper introduces Spider 2.0, a new NL2SQL benchmark designed to address the challenges of real-world enterprise text-to-SQL workflows. Spider 2.0 focuses on 595 complex workflow problems derived from real-world enterprise use cases involving databases often exceeding 1,000 columns and stored across diverse systems like BigQuery and Snowflake.

Overall, this paper is well structured and clearly written. The Spider 2.0 dataset is the first Text-to-SQL benchmark to incorporate complex contexts. The new dataset is publicly availbale and critical for the community and enterprises that develop LLM-based text-to-SQL solutions. Moreover, Spider 2.0 is also a valuable dataset for the future agent of data analytics. The benchmark creation is solid and clearly presented. The experimental analysis also provides some interesting findings.  All reviewers agree the acceptance of this paper.

**Additional Comments On Reviewer Discussion:**

In the rebuttal, the authors have clearly addressed the reviewers' questions regarding with the number of turns and the sequence to complete the task, the difficulty of the paper content, the ratio between simple syntax errors and completely incorrect chosen functions, the number of lines per SQL query, and others.

---

### Decision · Program_Chairs · 2025-01-22

Accept (Oral)